# Selective inhibition of cullin 3 neddylation through covalent targeting DCN1 protects mice from acetaminophen-induced liver toxicity

Haibin Zhou [1,7], Jianfeng Lu[1,7], Krishnapriya Chinnaswamy[2], Jeanne A. Stuckey[2], Liu Liu[1], Donna McEachern[1], Chao-Yie Yang [1], Denzil Bernard [1], Hong Shen [3], Liangyou Rui [3], Yi Sun [4] & Shaomeng Wang[1,5,6 ✉]

Cullin-RING E3 ligases (CRLs) regulate the turnover of approximately 20% of mammalian cellular proteins. Neddylation of individual cullin proteins is essential for the activation of each CRL. We report herein the discovery of DI-1548 and DI-1859 as two potent, selective and covalent DCN1 inhibitors. These inhibitors selectively inhibit neddylation of cullin 3 in cells at low nanomolar concentrations and are 2–3 orders of magnitude more potent than our previously reported reversible DCN1 inhibitor. Mass spectrometric analysis and co-crystal structures reveal that these compounds employ a unique mechanism of covalent bond formation with DCN1. DI-1859 induces a robust increase of NRF2 protein, a CRL3 substrate, in mouse liver and effectively protects mice from acetaminophen-induced liver damage. Taken together, this study demonstrates the therapeutic potential of selective inhibition of cullin neddylation.

[1] Department of Internal Medicine, University of Michigan, Ann Arbor, MI, USA. [2] Life Sciences Institute, University of Michigan, Ann Arbor, MI, USA. [3] Department of Molecular and Integrative Physiology, University of Michigan, Ann Arbor, MI, USA. [4] Institute of Translational Medicine, Zhejiang University, Hangzhou, Zhejiang, China. [5] Department of Pharmacology, University of Michigan, Ann Arbor, MI, USA. [6] Department of Medicinal Chemistry, University of Michigan, Ann Arbor, MI, USA. [7] These authors contributed equally: Haibin Zhou, Jianfeng Lu. ✉email: shaomeng@umich.edu

The ubiquitin-proteasome system (UPS) controls regulated degradation of intracellular proteins[1,2]. The UPS system has been successfully targeted for the development of drugs for the treatment of human diseases, as exemplified by the market approval of three proteasomal inhibitors for the treatment of multiple myeloma[3].

The cullin-RING ligases (CRLs) are the largest family of ubiquitin ligases, which regulate the turnover of ~20% of cellular proteins and play a key role in normal cellular physiology and various human diseases[4–7]. To date, eight CRL ligases have been identified in mammalian cells, which are characterized by the presence of eight individual cullin members[8,9]. Each CRL contains a common cullin-RING module as its catalytic core and has its own set of specific substrates[8,9].

The activities of all CRLs are tightly controlled by the neddylation process, which is initiated with the activation of NEDD8 (neuronal precursor cell-expressed developmentally down-regulated protein 8) by the E1 activating enzyme (NAE). This is followed by the transfer of the activated NEDD8 to one of the two NEDD8-specific E2 conjugating enzymes, UBC12 (UBE2M) and UBE2F, and then the activated NEDD8 is transferred from E2 to a cullin protein, leading to activation of individual CRLs[10–15].

CRLs have been pursued as potential therapeutic targets[8,9,16,17]. MLN4924 was discovered in 2009 as the first-in-class small-molecule inhibitor of the NAE, which effectively blocks activation of all CRLs[18,19]. MLN4924 is currently being evaluated in clinical trials for the treatment of human cancers[20,21]. Because each CRL has a different set of substrates, development of selective inhibitors for individual CRL members is of great interest but has proven to be challenging[8,9,16,17].

We recently reported our structure-based discovery of DI-591 (Fig. 1a, b) as a potent small-molecule inhibitor of the DCN1–UBC12 protein–protein interaction[22]. We demonstrated that DI-591 selectively inhibits the neddylation of cullin 3 over other cullin members[22]. Interestingly, in contrast to DI-591, other classes of reported DCN1 inhibitors showed no selective inhibition of the neddylation of cullin 3[22–30]. Although DI-591 is an excellent chemical biology tool compound to investigate cullin 3 in cells, its moderate cellular activity hampered its further testing in animal models of human diseases. Therefore, potent, selective, and in vivo active DCN1 inhibitors are needed to investigate the therapeutic potential of selective inhibition of the neddylation of cullin 3.

In the present study, we report our structure-based design of covalent DCN1 inhibitors based upon the cocrystal structure of the DI-591/DCN1 complex. Our efforts have led to the discovery of two potent DCN1 inhibitors, DI-1548 (**6**) and DI-1859 (**7**), both of which are 2–3 orders of magnitude more potent than DI-591 in selective inhibition of cullin 3 neddylation. Our mechanistic studies reveal their unique mechanism of fast covalent bond formation with DCN1. Employing these highly potent DCN1 inhibitors, we demonstrate that selective inhibition of cullin 3 neddylation effectively alleviates liver damage induced by acetaminophen in mice. Our study provides a proof of concept that selective inhibition of individual CRL members has a therapeutic potential for the treatment of human diseases.

## Results

**Structure-based design of highly potent covalent DCN1 inhibitors.** In addition to noncovalent DCN1 inhibitors, covalent DCN1 inhibitors have been reported[25]. We explored the possibility of designing a class of covalent DCN1 inhibitors based upon DI-591. Analysis of the cocrystal structure of DI-591 (**1**) in a complex with human DCN1 protein showed that the sulfur atom of Cys115 in DCN1 is located 4.1 Å from the α carbon of the 3-morpholinopropanamide unit in DI-591 (Fig. 1a), providing an opportunity for the design of covalent DCN1 inhibitors.

Based upon DI-591, we designed and synthesized compounds **2** and **3** (Fig. 1b), which contain chloroacetamide or acrylamide as a Michael acceptor group to form a covalent bond with the

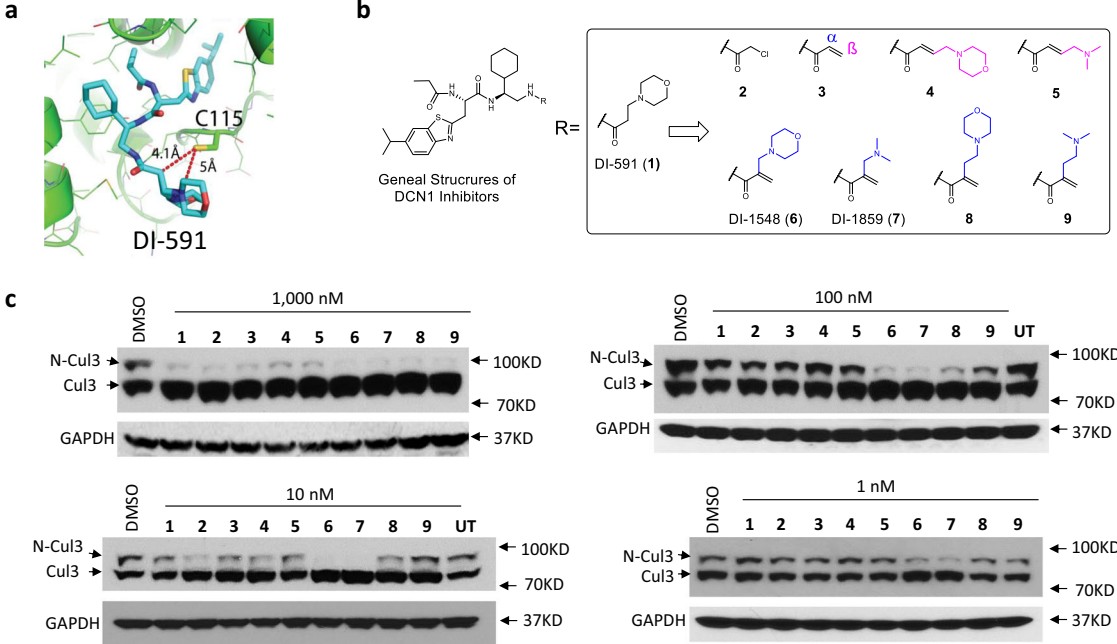

**Fig. 1 Structure-based design and initial cellular evaluation of covalent inhibitors of DCN1. a** Cocrystal structure of DCN1 (green) and DI-591 (cyan). **b** Chemical structures of designed and synthesized covalent inhibitors of DCN1. **c** Inhibition of cullin 3 neddylation by compounds **1**–**9**. Osteosarcoma U2OS cells were treated by nine DCN1 inhibitors at 1000, 100, 10, or 1 nM for 24 h. Protein levels of neddylated cullin 3 (N-Cul3) and unneddylated cullin 3 (Cul3) were examined by western blotting analysis with GAPDH as the loading control. Representative images of two independent experiments are shown and original images are provided as a Source Data file.

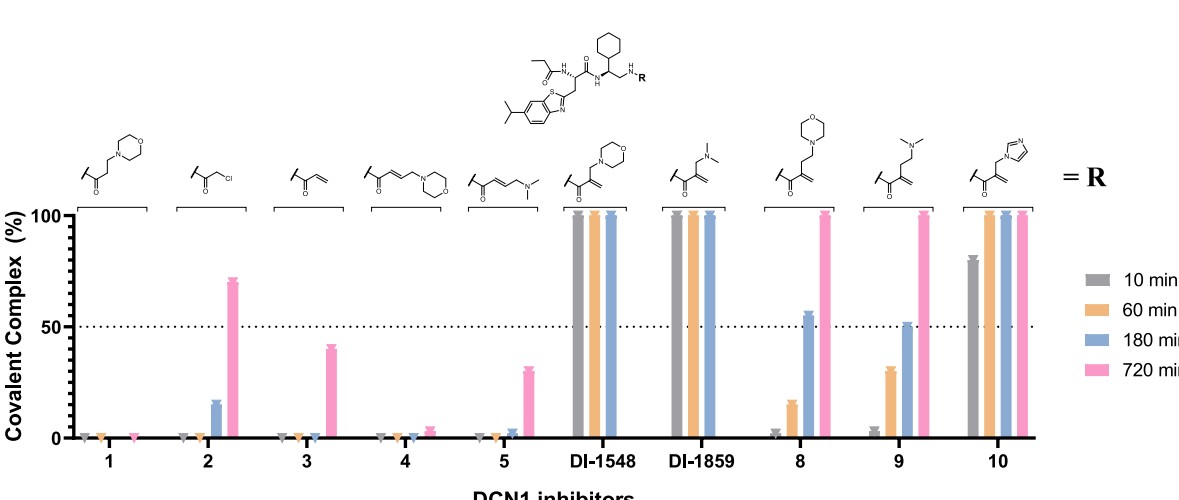

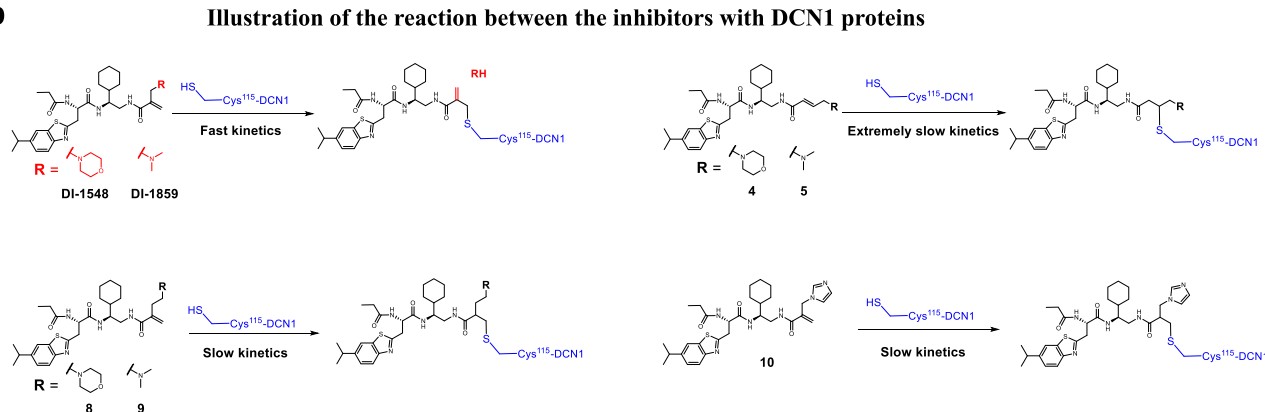

**Fig. 2 Mass spectrometric characterization of the reactivity of covalent DCN1 inhibitors with recombinant human DCN1. a** Percentage of the formation of covalent complex between the DCN1 protein and the indicated compounds based on the intact ESI mass spectrometric analyses at different time points. The mass spectra are provided in Supplementary Figs. 1–11. Source data are provided as a Source Data file. **b** Illustration of the reaction between the inhibitors with DCN1 protein based on mass spectrometry data.

Cys115 sulfur in DCN1. However, because both compounds were found to have poor solubility, we next synthesized a series of analogs by installation of a soluble group at either the α- or the β-carbon of the acrylamide group in **3** (Fig. 1b).

Since DI-591 was shown to selectively inhibit neddylation of cullin 3[22], we evaluated these potential covalent DCN1 inhibitors for their ability to inhibit neddylation of cullin 3 with DI-591 included as a control (Fig. 1c). While each of these compounds is effective at 1 μM in inhibition of cullin 3 neddylation, compounds DI-1548 (**6**), DI-1859 (**7**), **8**. and **9** are much more potent than DI-591. Specifically, DI-1548 and DI-1859 inhibit cullin 3 neddylation at concentrations as low as 1 nM and are thus ~1000 times more potent than DI-591.

**Mass spectral characterization of the reactivity of covalent and noncovalent DCN1 inhibitors with recombinant human DCN1 protein.** Although compounds **2–9** were all designed as covalent DCN1 inhibitors based upon DI-591 (**1**), their potencies in inhibition of cullin 3 neddylation differ by 2 orders of magnitude. We employed mass spectrometry to analyze the reactivity of compounds **2–9** with recombinant human DCN1 protein, with DI-591 (**1**) included as a control.

Consistent with its lack of a reactive Michael receptor group, DI-591 fails to forms a covalent complex with DCN1 protein (Fig. 2a and Supplementary Figs. 1 and 2). Compound **2**, which contains a reactive Michael acceptor group, shows no detectable complex formation with DCN1 with 1 h of incubation time but has 15 and 70% of complex formation with 3 and 12 h of incubation time, respectively (Fig. 2a and Supplementary Fig. 3). In comparison, compound **3** fails to react with DCN1 protein after 1 or 3 h incubation and reacts with 40% of DCN1 protein after 12 h of incubation (Fig. 2a and Supplementary Fig. 4). These data showed that while both **2** and **3** can indeed form a covalent complex with DCN1 protein, they exhibit slow reaction kinetics.

Installation of a soluble group onto the β-carbon of the acrylamide in **3** with either a morpholinyl or a dimethylamino group resulted in the crotonamide derivatives **4** and **5**, respectively (Fig. 1b). While compound **5** has a reaction kinetics similar to that of **3** in covalent complex formation with DCN1 protein, compound **4** has a slower reaction kinetics than **3** or **5** (Fig. 2 and Supplementary Figs. 5 and 6). Compound **4** has no detectable complex formation with DCN1 protein after 3 h incubation time and forms only 4% of the covalent complex after 12 h incubation.

Installation of the same two soluble groups onto the α-carbon of the acrylamide in **3** resulted in the methacrylamide derivatives

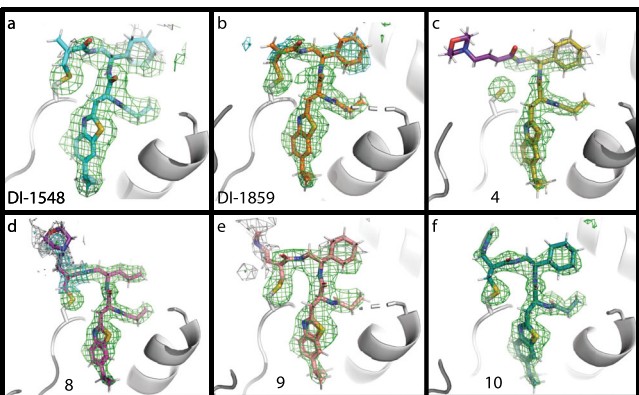

**Fig. 3 Cocrystal structures of six inhibitors bound to DCN1.** Omit electron density maps showing covalent binding of the compounds to DCN1 CYS Sγ (yellow), except for compound **4** whose omit map shows the lack of covalent bonding. The difference electron density map for each structure was calculated omitting the coordinates of the compound and CYS Sγ & Cβ atoms of DCN1. Contour levels for each map are **a** 2σ green, 1σ gray, **b** 3σ green, 2σ cyan, **c** 3σ green, 1.3σ gray, **d** 3σ green, 1.5σ cyan, 0.8σ gray, **e** 2.5σ green, 1σ gray, and **f** 3σ green. The ribbon of DCN1 backbone atoms are shown in gray, compounds are shown as sticks with nitrogen blue, oxygen red, sulfur yellow, and carbon for **a** DI-1548 (PDB ID: 6XOL), cyan, **b** DI-1859 (PDB ID: 6XOO), orange, **c** compound **4** (PDB ID: 6XOQ), gold, **d** compound **8** (PDB ID: 6XOM), magenta, **e** compound **9** (PDB ID: 6XON), pink, and **f** compound **10** (PDB ID: 6XOP), teal. All ligands were refined with atom occupancies of 1.0, except two groups shown with plum carbon atoms: methylmorpholine group of **8**, which was refined with occupancy of 0.5 and the allymorpholine group of compound **4**, which was refined with an occupancy of 0.

**6** (DI-1548) and **7** (DI-1859), respectively (Figs. 1b and 2a). In sharp contrast to compounds **4** and **5**, both DI-1548 and DI-1859 display very fast covalent complex formation with DCN1 protein (Fig. 2a and Supplementary Figs. 7 and 8). Within 10 min incubation, 100% of DCN1 protein has formed a complex with either DI-1548 or DI-1859. Interestingly, judging by the observed molecular weight for the complex, the morpholinyl group from DI-1548 or the dimethyl amine group from DI-1859 is cleaved upon complex formation (Fig. 2b and Supplementary Figs. 7 and 8).

To further explore the influence of the acrylamide substitution, we installed two soluble groups with one extra methylene group onto the α-carbon of the acrylamide in **3**, which yielded compounds **8** and **9**, respectively (Figs. 1b and 2a). While both **8** and **9** are able to form a covalent complex with DCN1 protein, they have a much slower reaction kinetics than **6** and **7** (Fig. 2a and Supplementary Figs. 9 and 10). For example, after 10 min incubation time, both **8** and **9** form only 2–3% of the DCN1-inhibitor complex and have ~50% of the DCN1-inhibitor complex formed after 3 h incubation (Fig. 2a and Supplementary Figs. 9 and 10). Unlike DI-1548 and DI-1859, there is no loss of either the morpholinyl from **8** or the dimethylamino group from **9** when they form a complex with DCN1 protein (Fig. 2b and Supplementary Figs. 9 and 10).

To further understand the fast kinetics of DI-1548 and DI-1859 with DCN1 protein, we synthesized compound **10** (Fig. 2b) by replacing the morpholinyl group in DI-1548 with an imidazole group. Compound **10** has a slower reaction kinetics with DCN1 than DI-1548 and DI-1859 (Fig. 2a and Supplementary Fig. 11). Furthermore, in contrast to DI-1548 and DI-1859, a covalent complex was formed between compound **10** and DCN1 protein without loss of the imidazole group in **10** based on mass spectrometry data (Fig. 2b and Supplementary Fig. 11).

Consistent with its slower reaction kinetics, compound **10** is ten times less potent than DI-1548 in cell in inhibition of cullin 3 neddylation (Supplementary Fig. 12).

Among those three cysteine residues (Cys90, Cys115, and Cys131) in DCN1 protein, Cys115 was predicted to form a covalent bond with these covalent inhibitors. To confirm this prediction, tandem mass spectrometry was performed to analyze the tryptic digested peptides from the unmodified DCN1 protein and DCN1 protein modified by DI-1548 (Supplementary Fig. 13). The tandem mass spectrometry data confirmed that DI-1548 indeed forms covalent bond with Cys115 in DCN1, accompanied with a loss of the morpholinyl group.

**Structural basis of the DCN1-inhibitor complex formation.** To gain precise structural insights into the DCN1-inhibitor complex formation, we determined six cocrystal structures of DCN1 in a complex with **6** (DI-1548), **7** (DI-1859), **4**, **8**, **9**, and **10** (Fig. 3 and Supplementary Table 1). As shown in Fig. 3a, b, the contiguous electron density between Cys115 in DCN1 and DI-1548/DI-1859 confirmed the covalent nature of these complexes. Consistent with the mass spectrometry data, the morpholinyl group in DI-1548 or the dimethylamino group in DI-1859 is absent in both cocrystal structures.

The cocrystal structure for compound **4** shows that there is no covalent bond formed between **4** and DCN1 protein (Fig. 3c), which is consistent with its very slow reaction kinetics in the mass spectral analysis (Fig. 2a). The cocrystal structure for compound **4** further shows that the Michael acceptor crotonamide in the compound and the sulfhydryl group of Cys115 in DCN1 are not in close proximity (3.9 Å) to initiate the covalent reaction, which explains the inactivity of **4** to DCN1 in covalent bond formation.

The cocrystal structures for compounds **8–10** with DCN1 reveal that these compounds all form a covalent bond with Cys115 residue of DCN1 (Fig. 3d–f). Consistent with the mass spectrometry data for these compounds and in contrast to that observed in the cocrystal structures for DI-1548 and DI-1859, their respective soluble group is not cleaved but retained in the complex structures for **8–10** (Fig. 3d–f).

As shown in these six cocrystal structures (Fig. 3), the common moieties in these compounds have similar interactions with DCN1 protein, when compared to those observed in the cocrystal structure for DI-591[22]. The benzothiazole ring of these compounds penetrates into a deep pocket of DCN1 and forms extensive hydrophobic interactions with DCN1. Additional hydrophobic contacts are also observed between DCN1 with the propionyl and cyclohexyl groups in these compounds. In addition, the propionyl group in these compounds forms two hydrogen bonds with DCN1. Therefore, with the exception of the covalent bond formation, these covalent compounds have very similar noncovalent interactions with DCN1 as compared to DI-591. Hence, the major difference in their potencies in inhibition of neddylation of cullin 3 in cell is determined by their reactivity of forming the covalent bond with DCN1.

**A proposed mechanism of covalent complex formation.** The fast reaction of DI-1548 and DI-1859 with DCN1, in comparison to the slow reaction of compounds **4**, **5**, **8**, and **9** with DCN1, suggests that the soluble group in DI-1548 or DI-1859 plays a major role in facilitating formation of the covalent bond between the Michael acceptor group in these compounds and Cys115 in DCN1. Based upon the mass spectrometric data and the cocrystal structures, we propose a concerted reaction mechanism for the formal $S_N2'$ substitution reaction between DCN1 with DI-1548 or with DI-1859 to explain the observed fast reaction kinetics for

**Fig. 4 A proposed mechanism of covalent complex formation between covalent DCN1 inhibitors and DCN1 protein.** The proposed mechanism of the formal $S_N2'$ substitution reaction between DI-1548 or DI-1859 and DCN1 protein, and the Michael addition reaction of DCN1 protein with **8** or **10**.

these two compounds (Fig. 4)[31–33]. In this proposed mechanism, the morpholine group of DI-1548 interacts with Cys115 in DCN1 and facilitates the proton transfer from Cys115 to morpholine with concurrent formation of the C–S bond between the ionized sulfur of Cys115 and the electron deficient allylic group of DI-1548. The morpholine group in DI-1548 thus has a dual role: first, the basic nitrogen interacts with Cys115, making the sulfur atom more acidic and facilitating covalent bond formation with the electron deficient C=C bond of the enone; second, formation of a new C=C bond is accompanied by breaking of the C–N bond, resulting in a loss of the morpholinyl group as the morpholine group is at the end of the allylic system. In this proposed mechanism, the intramolecular six-membered transition state for Michael addition and morpholine group dissociation are both important for the fast kinetics of DI-1548 in complex formation with DCN1. Similarly, the dimethyl amine group in DI-1859 functions in the same manner as the morpholine group in DI-1548 (Fig. 4).

For compound **8**, its basic nitrogen atom in its side chain can still interact with the Cys115 residue, making the sulfur atom more acidic and thus facilitating the covalent bond formation with the electron deficient allylic group (Fig. 4). However, the formation of the proposed seven-membered ring transition state for **8** is energetically less favorable than the formation of the proposed six-membered ring transition state for DI-1548 and DI-1859. Moreover, there is no cleavage of the morpholinyl group from **8** due to the extra methylene group (Fig. 4). As a result, the covalent complex formation between DCN1 and compound **8** is much slower than the covalent complex formation of DCN1 with either DI-1548 or with DI-1859.

Similar to **8**, while the imidazole group in **10** can be protonated to facilitate the C–S bond formation, it does not form an energetically favorable transition state and has no cleavage of its imidazole group (Fig. 4). Therefore, compound **10** is also less reactive than DI-1548 and DI-1859 in complex formation with DCN1. However, compound **10** still has a faster reactivity to DCN1 than compounds **8** and **9** probably due to the electron withdrawing effect of the protonated imidazole group.

**Determination of the specificity and irreversibility of DI-1548 and DI-1859 to DCN1.** To determine if a high binding affinity of these covalent inhibitors to DCN1 is required for their fast reactivity to the protein, we synthesized DI-1548DD, which is a stereoisomer of DI-1548 by inverting its two chiral centers (Fig. 5a). Binding experiments showed that DI-1548DD has a moderate binding affinity to DCN1 ($K_i = 1158$ nM), but it is >1000 times less potent than DI-1548 ($K_i < 1$ nM). Mass spectrometric analysis showed that in contrast to the rapid and complete formation of the DCN1/DI-1548 complex with 1 h incubation, DI-1548DD fails to form any detectable complex with DCN1 even after 3 h of incubation (Fig. 5b and Supplementary Figs. 14 and 15). These data show that the high binding affinity of DI-1548 to DCN1 is required for the covalent complex formation with DCN1 protein.

We investigated the specificity of DI-1548 and DI-1859 to DCN1 over DCN3. DCN3 contains six cysteine residues including Cys140 corresponding to Cys115 in DCN1, and DI-591 binds to DCN3 with a weak affinity ($K_d > 10$ μM)[22]. Mass spectrometric analyses showed that both DI-1548 and DI-1859 fail to form any detectable complex with DCN3 protein after 3 h

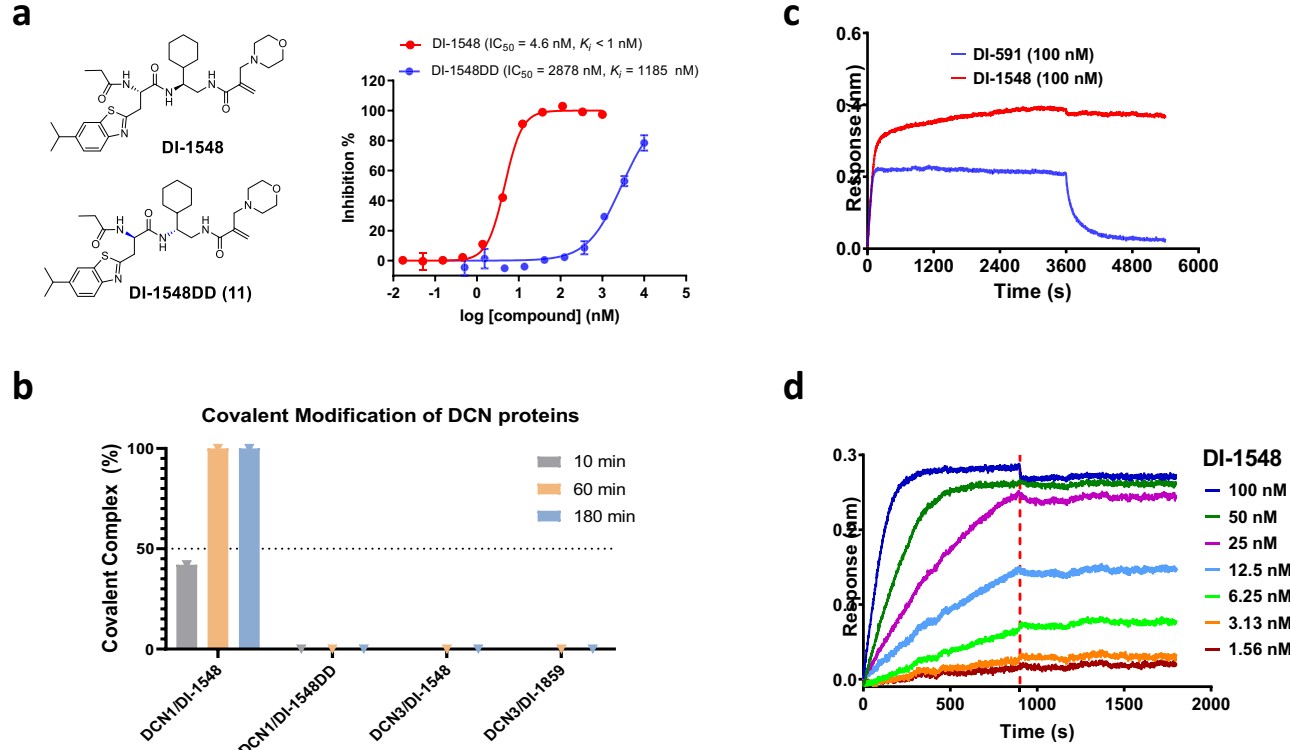

**Fig. 5 Characterization of the selectivity of DI-1548 and DI-1859. a** Chemical structures and competitive binding curves of DI-1548 and DI-1548DD.
**b** Percentage of the formation of covalent complex between the indicated compounds and proteins based on the intact ESI mass spectrometric analyses.
The mass spectra are provided in Supplementary Figs. 14–17. **c** Kinetic binding sensorgrams of DI-591 and DI-1548. Interferometric responses increased
upon the binding of compounds to the DCN1 protein immobilized on the super streptavidin (SSA) biosensor surfaces. Signals were monitored and recorded
every second while sample plates being continuously shaken at 1000 RPM to eliminate mass transport effect. **d** Kinetic binding sensorgrams of DI-1548
with concentrations ranging from 1.56 to 100 nM and steady state fitting of the equilibrium responses vs compound concentrations based on 1:1 binding
model. The representative data from at least two independent experiments were shown for **a**, **c**, and **d**. Source data are provided as a Source Data file.

of incubation (Fig. 5b and Supplementary Figs. 16 and 17),
highlighting their specificity.

We evaluated the reversibility of the DCN1/DI-1548 complex
formation by biolayer interferometry (BLI). The BLI data showed
that there is no dissociation phase for DI-1548 upon formation of
the complex, indicating that the complex formation is irreversible
(Fig. 5c). In contrast, a clear dissociation phase was observed for
the reversible inhibitor DI-591 (Fig. 5c). The interaction of DI-
1548 with DCN1 was investigated further by BLI, with the
inhibitor concentrations ranging from 1.56 to 100 nM. The BLI
signal was increased in a dose-dependent manner, reaching near
its maximum at 25 nM, and complete saturation at the two top
concentrations (Fig. 5d). During the wash-out phase of the
experiment, the signal remained constant at all concentrations,
further demonstrating that the formation of the inhibitor-protein
complex was irreversible.

**Determination of the cellular activity and selectivity of DI-
1548 and DI-1859 on inhibition of neddylation of cullin 3 and
other cullin members**. Our previous study showed that DI-591
effectively and selectively inhibits the neddylation of cullin 3 over
other cullin members[22]. We evaluated DI-1548 and DI-1859 for
their abilities to inhibit the neddylation of cullin 3 and other
cullin members in cells, with DI-591, MLN4924, DI-1548DD, and
DI-1859DD (**12**, a stereoisomer of DI-1859 by inverting its two
chiral centers) included as controls.

DI-1548 and DI-1859 are highly potent and effective in
inhibition of the neddylation of cullin 3 in cells with different
tissue types, such as osteosarcoma U2OS cell line (Fig. 6a),

immortalized THLE2 liver cells (Fig. 6b), breast cancer MDA-
MB-231 cell line (Supplementary Fig. 18), esophageal cancer
KYSE70 cell line (Supplementary Fig. 19), and colon cancer
HCT116 cell lines (Supplementary Fig. 20). DI-1548 and DI-1859
at concentrations as low as 0.3 nM significantly reduce the level of
neddylated cullin 3 and achieve profound inhibition of the
neddylation of cullin 3 at 1–3 nM (Fig. 6a, b and Supplementary
Figs. 18–21). DI-1859 is ~30 times more potent than MLN4924 in
inhibition of the neddylation of cullin 3 in THLE2 liver cells
(Fig. 6b and Supplementary Fig. 21). In direct comparison, DI-
1548 and DI-1859 are >300 times more potent than DI-591 in
different cell lines. DI-1548DD and DI-1859DD at 100–300 nM
have no significant effect on the neddylation of cullin 3 and are
thus at least >300 times less potent than DI-1548 and DI-1859,
respectively.

At concentrations up to 1000 nM, DI-1548 has no obvious
effect on the neddylation of other cullin members that were
examined, including cullin 1, 2, 4A, 4B, and 5 (Fig. 6a and
Supplementary Figs. 18–20). Hence, DI-1548 demonstrates
>1000-fold selectivity in inhibition of the neddylation of cullin
3 over other cullin members in a number of cell lines of different
human tissue types. As compared to DI-1548, DI-1859 has a
similar, high selectivity in inhibition of the neddylation of cullin 3
over other cullin members (Fig. 6b). As expected, MLN4294
potently inhibits neddylation of all cullins (Fig. 6b).

Our previous study[22] showed that DI-591 induces accumula-
tion of NRF2 protein, a substrate protein for CRL3 in cells.
Consistent with their high potency in inhibition of cullin 3
neddylation, DI-1548 and DI-1859, at concentrations as low as

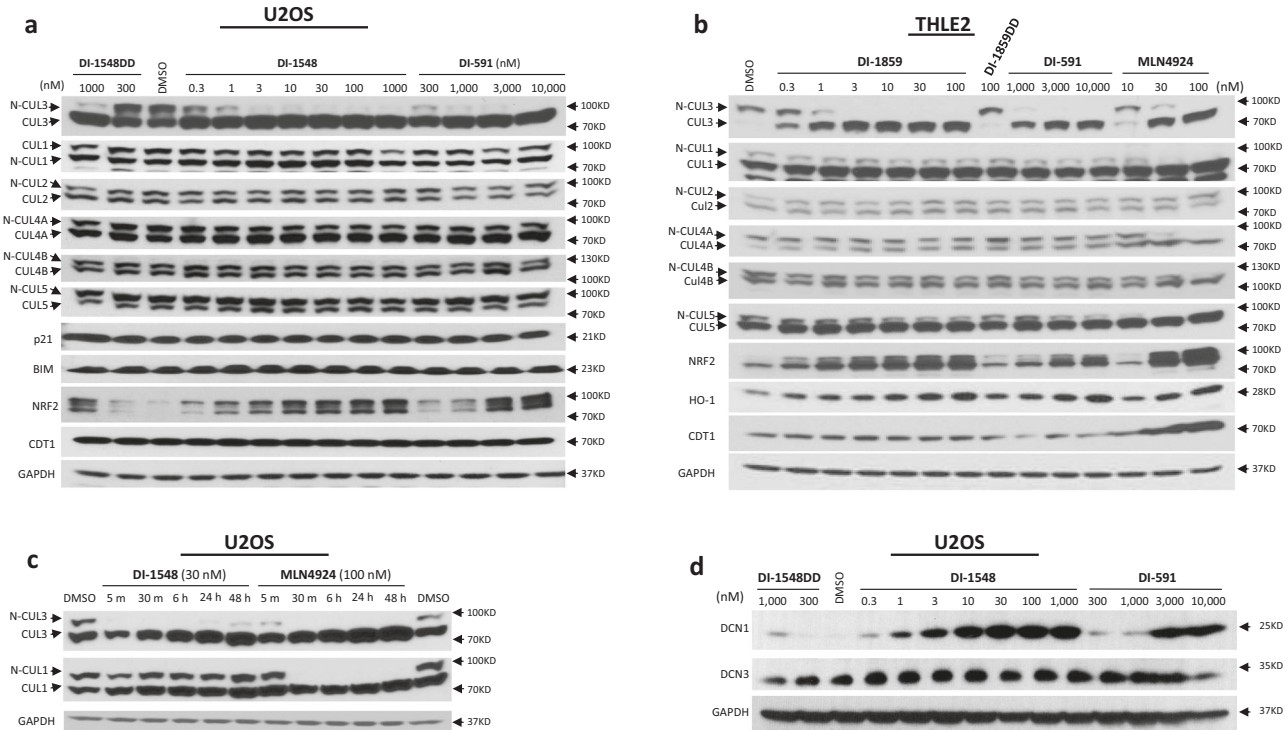

**Fig. 6 DI-1548 and DI-1859 selectively inhibit cullin 3 neddylation in different types of cells.** The effects of a dose range of the indicated compound on the protein level in osteosarcoma U2OS (**a**) and in immortalized THLE2 liver cell lines (**b**). The protein levels of neddylated cullin 3, 1, 2, 4A, 4B, and 5 (N-Cul3, 1, 2, 4A, 4B, and 5) and unneddylated cullin 3, 1, 2, 4A, 4B, and 5 (Cul3, 1, 2, 4A, 4B, and 5), as well as the substrates of cullins mediated CRLs, including p21, Bim, Nrf2 and CDT1, and HO-1 were examined by western blotting analysis. GAPDH was used as a loading control. **c** DI-1548 rapidly and selectively inhibits the neddylation of cullin 3 in osteosarcoma U2OS. The protein levels of neddylated cullin 3, 1 (N-Cul3, 1) and unneddylated cullin 3, 1 (Cul3, 1) were examined by western blotting analysis. GAPDH was used as a loading control. **d** DI-1548 effectively engages cellular DCN1 protein. U2OS cell line was treated by dose ranges of compounds DI-1548, DI-1548DD, or DI-591 for 1 h, then was heated at 53 °C. The cells were snap frozen in liquid nitrogen and thawed for three cycles. The levels of DCN1 and DCN3 in whole cell lysates were examined by western blotting analysis. GAPDH was used as a loading control. Representative images of two independent experiments are shown for **a**–**d**. Source data are provided as a Source Data file.

0.3–1 nM, induce significant accumulation of NRF2 protein (Fig. 6a, b and Supplementary Figs. 18–20) and are 100–1000 times more potent than DI-591. As NRF2 is negatively controlled by KEAP1 and p62 is able to activate NRF2 by disrupting the association of NRF2 and KEAP1[34], we examined the effect of DI-1548 and DI-1859 on KEAP1 and p62 proteins. DI-1548 and DI-1859 have no obvious effect on protein levels of KEAP1 and p62 in U2OS and THLE2 cells (Supplementary Fig. 22).

NRF2 is a transcriptional factor and regulates the expression of heme oxygenase-1 (HO-1) and NAD(P)H:quinone oxidoreductase (NQO1). Indeed, DI-1859 dose dependently upregulates the HO-1 protein (Fig. 6b) and markedly increases the mRNA levels of *HO-1* and *NQO1* (Supplementary Fig. 23). In contrast, no obvious accumulation of NRF2 protein occurs in cell treated with DI-1548DD (300 nM) or DI-1859DD (100 nM), consistent with their weak potency in inhibition of cullin 3 neddylation.

DI-1548 has no effect on the levels of p21 and BIM proteins, substrates of CRL1, and CDT1, a substrate of CRL4A, at concentrations up to 1000 nM, consistent with its inability to inhibit neddylation of these cullin members (Fig. 6a and Supplementary Figs. 18–20). DI-1859 also shows no effect on protein level of CDT1 (Fig. 6b).

Furthermore, DI-1548 shows no cytotoxicity in four cancer cell lines at concentrations up to 1000 nM, whereas MLN4294 potently inhibits cell viability in each of these four cancer cell lines (Supplementary Fig. 24). These data suggest that selective inhibition of cullin 3 neddylation has no significant effect on cell viability.

We examined inhibition kinetics by DI-1548 on cullin 3 neddylation, with MLN4924 included as the control (Fig. 6c). DI-1548 effectively inhibits the neddylation of cullin 3 within 5 min but has no effect on the neddylation of cullin 1 with treatment time up to 48 h. MLN4924 effectively inhibits the neddylation of both cullin 3 and 1 within 30 min. Hence, DI-1548 has fast kinetics in selective inhibition of the neddylation of cullin 3 over cullin 1.

To investigate if DI-1548 can effectively and potently interact with cellular DCN1 protein, we employed the cellular thermal shift assay (CETSA)[35] to examine the target engagement in U2OS cells. Our CETSA data showed that DI-1548 enhances the thermal stability of DCN1 protein in a dose-dependent manner and is effective at concentrations as low as 0.3 nM (Fig. 6d). In comparison, DI-1548 is >1000 times more potent than DI-591 and DI-1548DD in enhancing the thermal stability of cellular DCN1 protein (Fig. 6d). While DI-1548 is effective in enhancing the thermal stability of cellular DCN1 protein, it has no discernible effect on the thermal stability of cellular DCN3 protein at concentrations up to 1000 nM (Fig. 6d). Thus, our CETSA data provide evidence that DI-1548 effectively and potently interacts with cellular DCN1 protein.

**Evaluation of DI-1548 and DI-1859 in vivo**. We evaluated the pharmacodynamics (PD) of DI-1548 and DI-1859, our two most potent DCN1 inhibitors, in mice.

Neddylation of cullins is a dynamic process and detection of neddylation of cullins in tissues has proven to be very

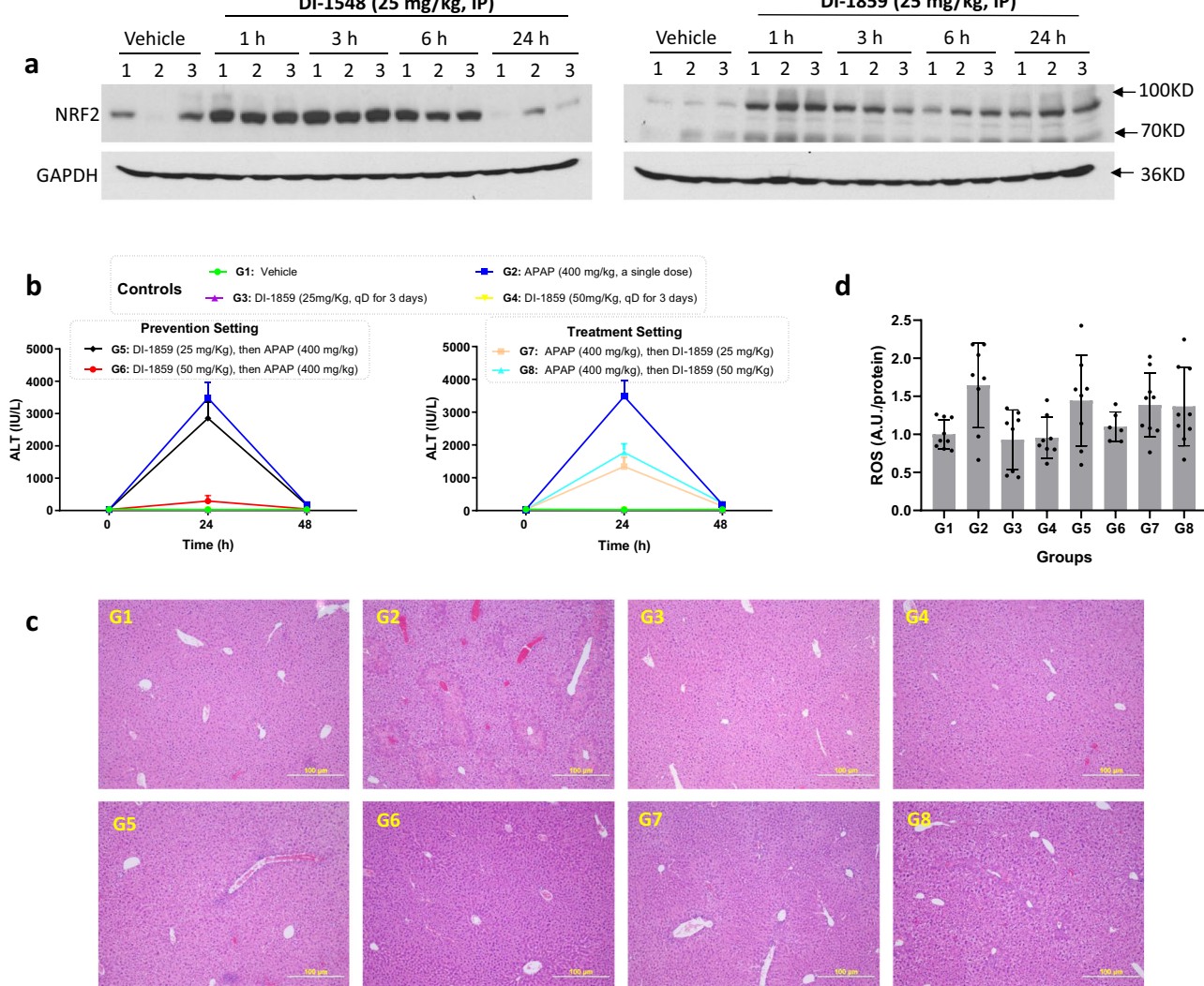

**Fig. 7 DI-1859 effectively increases the level of NRF2 protein in the liver tissue and protects liver tissue damage induced by acetaminophen (APAP) in mice. a** Western blotting analysis of the effect of DI-1548 or DI-1859 on the level of NRF2 protein in mouse liver tissue. Liver tissues harvested from C57BL/6 male mice treated with DI-1548 or DI-1859 at 25 mg/kg via intraperitoneal (IP) injection was lysed with RIAP buffer. The expression level of NRF2 protein was examined by western blotting. GAPDH was used a loading control. Representative images of two independent experiments are shown. **b** Analysis of the serum alanine aminotransferase (ALT) level in mice. The time (h) counted from APAP administration in G2 and G5-8, from DI-1859 or vehicle administration in G1, G3, and G4. Data are presented as mean ± s.d. with $n = 9$ (G1), 9 (G2), 8 (G3), 8 (G4), 9 G5), 9 (G6), 9 (G7), and 9 (G8). **c** Immunohistological analysis of mouse liver tissues. Images are representatives in each group. Images were taken with a 100× objective. Scale bar: 100 μm. **d** Analysis of the ROS level in mouse liver tissue. Data are presented as mean ± s.d. with $n = 9$ (G1), 8 (G2), 8 (G3), 8 (G4), 8 (G5), 6 (G6), 9 (G7), and 9 (G8). G1: Vehicle (qD for 3 days); G2: APAP (400 mg/kg, a single dose); G3: DI-1859 (25 mg/kg, qD for 3 days); G4: DI-1859 (50 mg/kg, qD for 3 days); G5: DI-1859 (25 mg/kg, qD for 3 days), a single dose of APAP at 400 mg/kg administered 3 h after the second dose of DI-1859 administration on day 2; G6: DI-1859 (50 mg/kg, qD for 3 days), a single dose of APAP at 400 mg/kg administered 3 h after the second dose of DI-1859 administration on day 2; G7: APAP (400 mg/kg, a single dose), followed with DI-859 at 25 mg/kg 3 h after APAP administration and two additional doses on day 2 and day 3; and G8: APAP (400 mg/kg, a single dose), followed with DI-859 at 50 mg/kg 3 h after APAP administration and two additional doses on day 2 and day 3. $n = 8$–9 mice per group. Source data are provided as a Source Data file.

challenging. Accordingly, we evaluated DI-1548 and DI-1859 for their ability to upregulate NRF2 protein in mouse liver tissue. Western blotting showed that a single dose of DI-1548 or DI-1859 effectively induces robust upregulation of NRF2 protein in the mouse liver (Fig. 7a). While the maximum upregulation of NRF2 by these two drugs is similar, DI-1859 produces an effect lasting longer than DI-1548, with upregulation of NRF2 persisting for >24 h (Fig. 7a). Of note, the PD effect for DI-1548 and DI-1859 in liver tissue extended after the compounds were cleared from systemic circulation (<3 h) based upon the pharmacokinetics analysis (Supplementary Table 2).

Based upon the promising PD data, we further investigated DI-1859 in mice.

We evaluated DI-1859 for its potential toxicity in immuno-competent mice with daily, 50 mg/kg, intraperitoneal (IP) dosing for 14 consecutive days (Supplementary Table 3). Overall, administration of DI-1859 at 50 mg/kg for 14 days was well tolerated. There were no changes in body weights (Supplementary Fig. 25) or organ weights (Supplementary Table 4), and no meaningful alterations in hematologic parameters (Supplementary Table 5). Histological analysis showed that after 14-day daily treatment, there was no observable difference between vehicle-

and DI-1859-treated mice in all the tissues examined, including liver, heart, kidney, lung, intestine, stomach, pancreas, colon, and spleen (Supplementary Table 6). These results indicate that DI-1859 shows no toxicity in mice with repeated, daily administration for 14 days.

NRF2 activators have been studied for alleviation of tissue damage mediated by oxidative stress[36,37] and we therefore investigated the potential protective effect of DI-1589-mediated NRF2 upregulation on normal tissues in mice. In the clinic, overdosing with acetaminophen (APAP or Tylenol) is responsible for more than 50% of overdose-related acute liver failure cases and ~20% of the liver transplant cases in the USA[38]. Since DI-1859 effectively upregulated NRF2 in liver tissue in mice, we evaluated DI-1859 for its ability to mitigate APAP-induced hepatotoxicity in mice with the data summarized in Fig. 7b–d.

A single, large dose of APAP (400 mg/kg, IP) induced severe hepatotoxicity, as shown by a substantial increase of alanine aminotransferase (ALT) by at least two orders of magnitude (Fig. 7b), and necrosis and bleeding in hepatic centrilobular area in liver tissue observed in histology analysis (Fig. 7c). In comparison, DI-1859 at both 25 and 50 mg/kg caused no liver tissue damage (Fig. 7b, c). In the prevention setting (G5 and G6) in which DI-1589 was administered prior APAP in mice, DI-1859 at 25 and 50 mg/kg reduced the elevated ALT level induced by 400 mg/kg of APAP by 20%, and 90%, respectively, at the 24 h time point (Fig. 7b). While DI-1859 at 25 mg/kg largely abolished tissue distortion and greatly reduced acetaminophen-induced tissue damage, DI-1859 at 50 mg/kg near completely prevented APAP-induced liver tissue damage (Fig. 7c). In the treatment setting (G7 and G8) in which DI-1589 was administered after APAP administration in mice, DI-1859 at both 25 and 50 mg/kg reduced the APAP-induced ALT elevations by ~50% (Fig. 7b) and also significantly reduced the liver damage induced by APAP (Fig. 7c).

Consistent with the observed ALT levels (Fig. 7b) and immunohistological data (Fig. 7c) in different groups, DI-1859 also effectively reduced the level of reactive oxygen species (ROS), in both prevention and treatment settings, with 50 mg/kg of DI-1859 in the prevention setting being the most effective (Fig. 7d).

Collectively, these in vivo data demonstrate that DI-1859 is very effective in reducing the acetaminophen-induced liver tissue damage in both pretreatment and posttreatment settings.

## Discussion

CRLs regulate the turnover of ~20% of mammalian cellular proteins and constitute a large number of potential therapeutic targets. Although protein substrates regulated by each of the eight individual CRLs have not been fully identified, it is clear that each CRL targets a distinctive set of proteins for degradation[8,9]. Accordingly, it is very valuable to develop potent and selective small-molecule inhibitors for individual CRL members in order to further define their roles in different biological processes and human diseases, as well as to explore therapeutic applications of selectively targeting individual CRLs.

Our recent discovery of DI-591 as a selective small-molecule inhibitor of cullin 3 neddylation through targeting the DCN1–UBC12 protein–protein interaction has provided an exciting opportunity to explore the potential therapeutic applications of selective inhibition of the neddylation cullin 3. Unfortunately, DI-591 has only moderate cellular potency, which hampered further testing of its therapeutic potential in vivo. To address this major limitation, we have designed a class of covalent small-molecule inhibitors based upon the cocrystal structure of DI-591 complexed DCN1.

Our efforts have led to the discovery of the methacrylamide-type compounds, DI-1548 and DI-1859, as two potent irreversible DCN1 inhibitors. DI-1548 and DI-1859 are able to react rapidly with DCN1, while their close analogs, the crotonamide-type compounds (**4** and **5**) have low reactivity to DCN1 (Fig. 2). Our mass spectrometry analysis and determination of cocrystal structures revealed that DI-1548 and DI-1859 achieve their high potency in targeting DCN1 through a unique reaction mechanism in which a covalent bond is formed between the inhibitors and the Cys115 residue in DCN1, accompanied with cleavage of a soluble group in the inhibitors. Although DCN3 have a Cys140 residue in the binding site corresponding to Cys115 in DCN1, DI-1548 and DI-1859 fail to react with DCN3. These data clearly reveal that the high binding affinity to DCN1, the orientation of acrylamide, the proposed intramolecular six-membered transition state for the Michael addition, and the cleavage of the soluble group are all critical for the fast kinetics of DI-1548 and DI-1859 in complex formation with DCN1.

DI-1548 and DI-1859 effectively inhibit neddylation of cullin 3 at concentrations as low as 0.3 nM and are ~1000 times more potent than DI-591. Of significance, DI-1548 and DI-1859 demonstrate 1000 times selectivity in inhibition of the neddylation of cullin 3 over all other cullin members examined in cell lines of different tissue types. Consistent with the greatly improved potency in inhibition of cullin 3 neddylation, DI-1548 and DI-1859 effectively induce upregulation of NRF2, a substrate of CRL3, at concentrations as low as 0.3 nM. Furthermore, DI-1548 and/or DI-1859 have no effect on p21 and BIM proteins, substrates of CRL1, and CDT1, a substrate of CRL4A at concentrations 1000 times higher than the minimal effective concentration in inducing upregulation of NRF2. Consistently, our CTSA experiment showed that DI-1548 effectively stabilizes cellular DCN1 protein at concentrations as low as 0.3 nM. Collectively, our data demonstrate that DI-1548 and DI-1859 are highly potent and selective in inhibition of cullin 3 neddylation via covalent targeting DCN1. Importantly, the selective inhibition of neddylation of cullin 3 causes no cytotoxicity in cells.

We next performed critical in vivo experiments to assess the therapeutic potential of DI-1548 and DI-1859 in mice. We inquired if a single administration of DI-1548 and DI-1859 is effective in inhibition of cullin 3 neddylation in vivo. Because neddylation is a dynamic process in tissue, determination of the neddylated cullin 3 has proven to be very challenging. Therefore, we examined upregulation of NRF2 protein as a surrogate marker for inhibition of cullin 3 neddylation. Our PD data clearly showed that a single administration of both DI-1548 and DI-1859 is very effective in inducing upregulation of NRF2 protein in liver tissue in mice with DI-1859 being more effective and longer lasting than DI-1548. We thus investigated DI-1859 further for its potential toxicity and therapeutic applications in mice.

Although covalent small-molecule inhibitors have been successfully developed for the treatment of human diseases, including cancer, there is often a concern of their potential toxicity from possible off-target reactivity[39,40]. We have demonstrated that DI-1548 and DI-1859 selectively and rapidly react to Cys115 in DCN1 and show no reactivity to DCN3 despite the fact that DCN3 has a Cys140 in its binding site corresponding to Cys115 in DCN1. To further address the general toxicity concern associated with a covalent inhibitor in vivo, we evaluated the potential toxicity of DI-1859 in mice with daily dosing at 50 mg/kg for 14 consecutive days. Our extensive histology analysis on a large number of tissues, including liver, lung, kidney, and heart, showed that DI-1859 causes no tissue damage in any tissue in mice, indicating that DI-1859 has very low toxicity in mice.

A potential major advantage of covalent inhibition is the persistent effect on its biological target as compared to noncovalent

inhibition[41]. Indeed, DI-1548 and DI-1859 not only achieve >300 times better cellular potency than their noncovalent counterparts but also deliver persistent PD effect in liver tissue, even after the drugs are cleared from plasma and liver tissue.

To date, DCN1 inhibitors have not been evaluated in any therapeutic setting in vivo. Previous studies have suggested that upregulation of NRF2 can protect tissues from damage induced by oxidative stress[36,37]. Because DI-1859 effectively upregulates NRF2 in liver tissue in mice, we tested its protective effect of normal liver tissue in mice from profound oxidative damage induced by a high dose of acetaminophen, which is responsible for 20% of liver transplants in the USA[38]. Our data showed that a high dose of acetaminophen (400 mg/kg, IP) induced profound liver damage, as indicated by the highly elevated level of a liver enzyme ALT and ROS, as well as based upon histology analysis of the liver tissue (Fig. 7b–d). When DI-1859 was administered prior to a high dose of acetaminophen, it effectively protected liver damage induced by acetaminophen. DI-1859 at 50 mg/kg reduced the dramatically elevated level of ALT induced by acetaminophen by >90% at the 24 h time point. Even when DI-1859 was administered after the dosing of acetaminophen at which time point the liver tissue damage had already occurred, DI-1859 was still effective in reducing the dramatically elevated level of ALT by 50% with both 25 and 50 mg/kg doses. Analysis of the ROS level, as well as histology analysis of the liver tissue confirmed that DI-1859 effectively reduced the increased ROS level and tissue damage by APAP. Taken together, our in vivo data provide strong evidence that DI-1859 may have a therapeutic potential in protection of normal tissues from oxidative stress induced by APAP or potentially other drugs or agents.

Figure 8 illustrates our proposed mechanism of action for covalent DCN1 inhibitors. DCN1 functions as a scaffold protein and upholds the neddylation complex (RBX1–UBC12 ~ NEDD8–CUL3–DCN1) through interacting with both UBC12 and cullin 3 proteins. Upon neddylation, cullin 3 undergoes a conformational change to promote the assembly and activation of the CRL3 complex (CUL3–RBX1–KEAP1)[42]. This activated CRL3 complex recruits substrate proteins, such as NRF2, leading to their ubiquitination, followed by degradation by the 26S proteasome.

Covalent inhibitors such as DI-1548 and DI-1859 form a covalent bond with DCN1 and disrupt the interaction of DCN1 with UBC12, leading to dissociation and inactivation of the CRL3 complex. Inactivation of the CRL3 complex results in accumulation of CRL3 substrate proteins, such as NRF2, in cells and tissues. Accumulated NFR2 protein is translocated into the nucleus and regulates its targeted genes including *HO-1* and *NQO1*, which protect mice from liver damage induced by acetaminophen.

Also illustrated in Fig. 8, KEAP1 functions as the substrate adapter in the CRL3 complex through interacting with cullin 3 and NRF2. Hence, small-molecule inhibitors of the KEAP1–NRF2 interaction are also capable of inducing the accumulation of NRF2[34,43]. Thus, pharmacological activation of NFR2 can be achieved by both DCN1/UBC12 inhibitors and KEAP1/NRF2 inhibitors. Our data indicate that DCN1/UBC12 inhibitors selectively inhibit the neddylation of cullin 3 and subsequently inactivate CRL3, which induces the accumulation of CRL3 substrates including but not limited to NRF2. In comparison, KEAP1/NRF2 inhibitors block the interactions of KEAP1 with NRF2 or potentially other KEAP1 interacting proteins, which use the same docking site, leading to upregulations of those proteins recruited by KEAP1 to CRL3 for ubiquitination. Additional studies are required to understand the similarities and differences on regulation of substrate proteins between DCN1/UBC12 inhibitors and KEAP1/NRF2 inhibitors.

In summary, our present study reports the discovery of DI-1548 and DI-1859 as two potent, covalent small-molecule inhibitors of DCN1, both of which potently and selectively inhibit the neddylation of cullin 3 over other cullin members. We have shown that selective inhibition of cullin 3 is effective in protecting mice from liver damage induced by acetaminophen. Hence, our study suggests that selective inhibition of individual CRLs may have a therapeutic potential for the treatment of different human diseases and conditions.

## Methods

**Chemistry**. The chemistry methods including synthetic procedures and characterization data for all those DCN1 inhibitors are provided in Supplementary Methods.

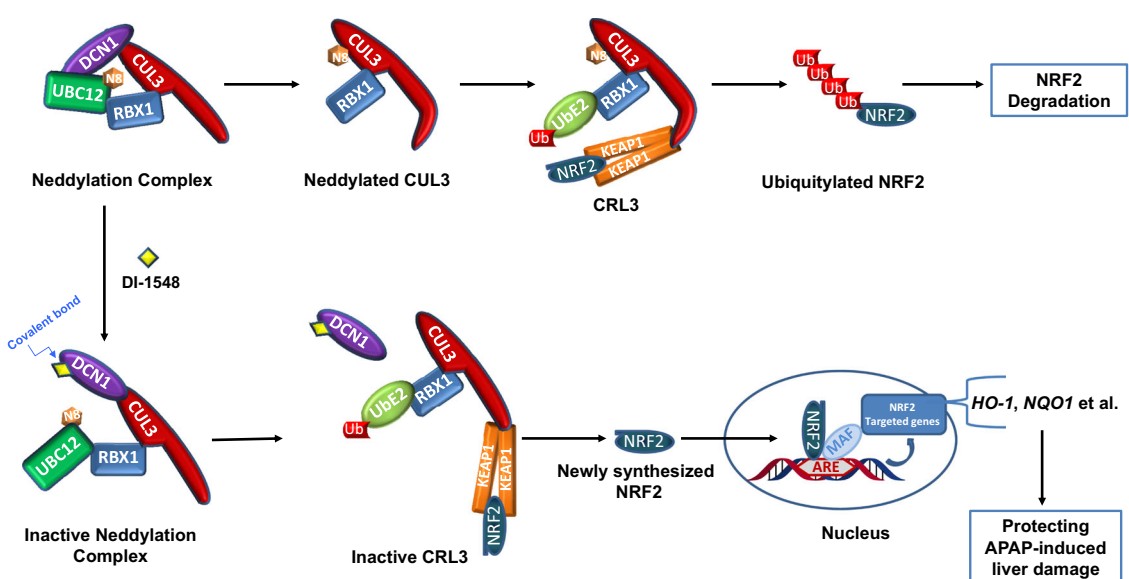

**Fig. 8 Proposed mechanism of action of covalent DCN1 inhibitor DI-1548 for its protection of APAP-induced liver damage via selective inhibition of cullin 3 neddylation.** DI-1548 covalently binds to DCN1 and disrupts the interaction of DCN1–UBC12 and causes the collapse of neddylation complex. The resultant inactive CRL3 leads to the accumulation of NRF2 which upregulates its targeted genes and effectively protects mice from acetaminophen-induced liver damage.

**Western blot analysis**. Western blotting was performed as described previously[22]. Treated cells were lysed by RIPA buffer supplemented with protease inhibitor. The expression level of indicated proteins was examined by immunoblotting analysis and GAPDH was used as the loading control. Antibodies were purchased commercially: cullin 1 (sc-17775, 1:500 dilution) and cullin 2 (sc-166506, 1:500 dilution) from Santa Cruz Biotech (Santa Cruz, CA); cullin 4A (PA5-14542, 1:1000 dilution), cullin 4B (PA5-50647, 1:1000 dilution), and DCN3 (DCUN1D3, PA5-44000, 1:1000 dilution) from Thermo Fisher Scientific (Wayne, MI); cullin 3 (2759, 1:1000), NRF2 (12721, 1:1000 dilution), HO-1 (70081, 1:1000 dilution), and Bim (2819, 1:1000 dilution) from Cell Signaling Technology (Boston, MA); DCN1 (GWB-E3D700, 1:1000 dilution) from GenWay Biotech (San Diego, CA); and cullin 5 (A302-173A, 1:500 dilution) from Bethyl Labs (Montgomery, TX). Results are representative of two independent experiments.

**Mass spectrometric analysis of DCN1 protein incubated with DCN1 inhibitors**. Method A: Recombinant human DCN1 protein (380 μM in 25 mM Tris 7.5, 200 mM NaCl, 1 mM DTT buffer) was incubated with 1.2-fold excess of a DCN1 inhibitor at 37 °C for 10 min, 1 h, 3 h, or 12 h. Following incubation, the protein was diluted with water to 1 mg/mL, and the protein was analyzed by LC–MS (Agilent Q-TOF 6545) under the following conditions: fragmentor voltage, 300 V; skimmer voltage, 75 V; nozzle voltage, 100 V; sheath gas temperature, 350 °C; and drying gas temperature, 325 °C. MassHunter Qualitative Analysis software (Agilent) was used to analyze the data. Intact protein masses were obtained using the maximum entropy deconvolution algorithm. The mass spectrometry data in Supplementary Figs. 1–11 were generated by Method A.

Method B: Recombinant DCN1 or DCN3 protein (100 nM in in 25 mM Tris 7.5, 200 mM NaCl, 1 mM DTT buffer) was incubated with 1.2-fold excess of the DCN1 inhibitors at 37 °C for 10 min, 1 h, or 3 h. Following incubation, the protein was analyzed by LC–MS (Agilent Q-TOF 6545) under the following conditions: fragmentor voltage, 300 V; skimmer voltage, 75 V; nozzle voltage, 100 V; sheath gas temperature, 350 °C; and drying gas temperature, 325 °C. MassHunter Qualitative Analysis software (Agilent) was used to analyze the data. Intact protein masses were obtained using the maximum entropy deconvolution algorithm. The mass spectrometry data in Supplementary Figs. 14–17 were generated by Method B.

**High-resolution LC–tandem mass spectrometry**. Equal amounts of proteins (10 μg) were denatured in 8 M urea. Cysteines were reduced with 10 mM DTT at 45 °C for 30 min and alkylated using 50 mM chloroacetamide (RT, 30 min). Upon dilution of urea to <1.2 M, digestion with 250 ng of sequencing grade, modified trypsin (Promega) was carried out overnight at 37 °C. Reaction was terminated by acidification with trifluoroacetic acid (0.1% v/v) and peptides were purified using SepPak C18 cartridge following the manufacturer's protocol (Waters Corp). An aliquot of the resulting peptides (~2 μg) was resolved on a nanocapillary reverse phase column (Acclaim PepMap C18, 2 μm, 50 cm, Thermo Fisher Scientific) using 0.1% formic acid/acetonitrile gradient at 300 nL/min and directly introduced in to Q Exactive HF mass spectrometer (Thermo Fisher Scientific, San Jose, CA). MS1 scans were acquired at 60 K resolution (AGC target = 3e6, max IT = 50 ms). Data-dependent high-energy C-trap dissociation MS/MS spectra were acquired for the 20 most abundant ions (Top20) following each MS1 scan (15 K resolution; AGC target = 1e5; relative CE ~ 28%). Proteins were identified by searching the data against restricted database containing DCN1 protein sequence and common contaminants using Sequest HT (Proteome Discoverer v2.4, Thermo Fisher Scientific). Search parameters included MS1 mass tolerance of 10 ppm and fragment tolerance of 0.02 Da; two missed cleavages were allowed; carbamidomethylation (+57.02146 Da) and DI-1548 adduct (+510.2665 Da) of cysteine, and oxidation of methionine, deamidation of asparagine and glutamine were considered as potential modifications. False discovery rate (FDR) was determined using target decoy validator and proteins/peptides with an FDR of ≤1% were retained for further analysis. The DI-1548 modified peptide PSMs were manually verified.

**Purification of lysozyme–DCN1 chimera protein**. Lysozyme–DCN1 gene was synthesized fusing lysozyme residues 1–164, containing mutations (C54T, C97A, A146T, D127A, and R154A), to DCN1 residues 62–251. Ligation independent cloning was utilized to clone the fused gene into an N-terminal His6–TEV expression vector. The clone was transformed into Rosetta 2 cells, grown in terrific broth at 37 °C, and expressed overnight with 0.4 mM isopropyl β-D-1-thiogalactopyranoside at 20 °C. Pelleted cells were freeze-thawed and lysed in 25 mM HEPES, pH 7.5, 150 mM NaCl, 0.1% β-mercaptoethanol and protease inhibitors. Cellular debris was removed via centrifugation. The supernatant was loaded onto a Ni-NTA (Qiagen) column at 4 °C. The column was washed with 25 mM HEPES, pH 7.5, 150 mM NaCl, and 20 mM imidazole, then protein eluted with buffer containing 300 mM imidazole. The eluate was applied to a Superdex 200 (GE Healthcare) equilibrated with 25 mM HEPES, pH 7.5, 200 mM NaCl, and 1 mM DTT. The protein was judged to be greater than 95% pure by SDS-PAGE, concentrated to 3.5 mg/mL and stored at −80 °C. Yield was 450 mg/L of expressed media.

**Crystallization and structure determination of lyso–DCN1-inhibitor complexes**. Lyso–DCN1 protein was thawed and concentrated to 9.9 mg/mL in 25 mM

HEPES, pH 7.5, 200 mM NaCl, and 1 mM DTT. Each compound was incubated with protein in a 1:1.2 protein to compound molar ratio overnight at 4 °C, then spun and setup in sitting drop vapor diffusion experiments. Crystallization drops contained 2 μL of protein–inhibitor solution and 1 μL of well solution. Crystals grew at 20 °C with well solution containing 28–33% PEG 3350 and 200 mM ammonium formate. Crystals were cryoprotected in 25 mM HEPES, pH 7.5, 200 mM NaCl, 30% PEG 3350, and 1 mM DTT.

Diffraction data were collected at a wavelength of 0.9786 Å on the APS LS-CAT 21-ID-G beamline at Argonne National Laboratory. Data were processed using HKL2000[44] and solved via molecular replacement using molrep[45] with two independent search models (lysozyme and DCN1) from PDB ID 5V83. The structures were iteratively fit and refined using Coot[46] and Buster[47]. The coordinates and restraints for the inhibitors were derived from Grade using the mogul+qm option[47]. Each structure was solved with one protein–inhibitor complex per asymmetric unit. Structures were validated using Molprobity[48] and the PDB validation server. Residue numbering used in this report is for DCN1 alone. DCN1 residue numbering for the deposited coordinates are +1000. All inhibitors covalently bound to Cys115 with the exception of compound **4**. Data collection and refinement statistics are listed in Supplementary Table 1.

**Competitive FP binding assay**. Competitive FP binding assay was performed as previously described[22]. The IC50 and $K_i$ values of DI-1548 and DI-1548DD were determined in competitive binding experiments. Mixtures of preincubated protein/probe complex solution in the assay buffer (196 μL) and compounds in DMSO (4 μL) were added into assay plates which were incubated at room temperature for 30 min with gentle shaking. The concentrations of DCN1 protein and fluorescent probe were 50 and 5 nM, respectively. Positive controls containing only free probes (equivalent to 100% inhibition) and negative controls containing protein/probe complex only (equivalent to 0% inhibition) were included in each assay plate. FP values in millipolarization units were measured using the Infinite M-1000 plate reader (Tecan U.S., Research Triangle Park, NC) in Microfluor 1 96-well, black, round-bottom plates (Thermo Fisher Scientific, Waltham, MA) at an excitation wavelength of 485 nm and an emission wavelength of 530 nm. IC50 values were determined by nonlinear regression fitting of the competition curves using the dose–response inhibition equation (four parameters, variable slopes) included in the Prism/GraphPad software. $K_i$ values were calculated using methods described previously[49].

**BLI assay**. BLI experiments were performed using an OctetRED96 instrument from PALL/ForteBio, as previously described[22]. All assays were run at 30 °C using PBS (pH 7.4) as the assay buffer, in which 0.1% BSA, 0.01% Tween-20, and 2% DMSO were added. DCN1 protein was biotinylated using the Thermo EZ-Link long-chain biotinylation reagent. Biotinylated DCN1 protein was tethered on super streptavidin biosensors (ForteBio) by dipping sensors into 10 μg/mL protein solutions. To eliminate loose nonspecific bound protein and establish a stable base line, sensors with proteins were moved and dipped into wells with pure assay buffer and equilibrated in the buffer for 10 min. Association–dissociation cycles were performed by moving and dipping sensors into compound solution wells then into pure buffer wells. DMSO-only reference was included in all assays. Raw kinetic data collected were processed with the data analysis software provided by the manufacturer using double reference subtraction in which both DMSO-only reference and inactive protein reference were subtracted.

**Cell viability assay**. Cell viability assay was performed as previously described[22]. The effect of DCN1 inhibitors on cell growth was evaluated by a WST-8 [2-(2-methoxy-4-nitrophenyl)-3-(4-nitrophenyl)-5-(2,4-disulfophenyl)-2H-tetrazolium, monosodium salt assay (Dojindo Molecular Technologies, Inc). Cells (2000–3000 cells in each well) were seeded in 96-well tissue culture plates in medium (100 μL) and treated with serial dilution of DCN1 inhibitors for 3 days. At the end of incubation, WST-8 dye (20 μL) was added to each well and incubated for 1–3 h, then the absorbance was measured in a microplate reader (Molecular Devices) at 450 nm. Cell growth inhibition was evaluated as the ratio of the absorbance of the sample to that of the control. The values were inserted into Prism software, and the curves and the EC50 values were produced by Prism software.

**Cellular thermal shift assay**. CETSA was performed as previously described[22]. The U2OS cells (5 × 10^5 per sample) were treated with compounds or with DMSO for 1 h, washed with PBS for three times, and dissolved in 50 μL PBS supplemented with protease inhibitor, followed by heating at 53 °C in a Mastercycler gradient (Eppendorf, New York, USA). Treated cells were then subjected to snap-freezing in liquid nitrogen and thawed on ice for three cycles. The protein level of DCN1 and DCN3 in equal amount of the supernatant was examined by western blot methods. GAPDH was used as a control. Results are representative of three independent experiments.

**APAP-induced liver injury**. The in vivo studies were performed under animal protocols (PRO0007499) and (PRO00006638) approved by the Institutional Animal Care & Use Committee of the University of Michigan, in accordance with

the recommendations in the Guide for the Care and Use of Laboratory Animals of the National Institutes of Health. C57BL/6 WT mice (~8 weeks, male, normal chow) were purchased from the Jackson Laboratory and randomly divided into eight groups (8–9 mice per group) treated (IP) as follows: phosphate-buffered saline vehicle once per day for 3 days (G1), APAP alone a single dose at 400 mg/kg on day 1 (G2), DI-1859 alone 25 mg/kg once per day for 3 days (G3), DI-1859 alone 50 mg/kg once per day for 3 days (G4), pretreatment (1 day before the APAP administration) with DI-1859 at 25 mg/kg once per day for 3 days followed with APAP at 400 mg/kg on day (G5), pretreatment (1 day before the APAP administration) with DI-1859 at 50 mg/kg once per day for 3 days followed with APAP at 400 mg/kg on day (G6), APAP a single dose at 400 mg/kg (3 h before the first DI-1859 administration) followed with DI-1859 at 25 mg/kg once per day for 3 days (G7), and APAP a single dose at 400 mg/kg (3 h before the first DI-1859 administration) followed with DI-1859 at 50 mg/Kg once per day for 3 days (G8).

To induce acute liver injury, animals received PBS or APAP (400 mg/kg, sigma, A7085) through I.P. injection at 12 p.m. and were sacrificed 48 h afterward. To examine the protective effect of DI-1859 (groups 5 and 6), animals were I.P. injected with compound for 3 consecutive days (9 a.m. on each day, starting 1 day before APAP administration). To examine the restorative effect of DI-1859 (groups 7 and 8), animals were I.P. injected with compound 3 h after APAP administration followed by two additional treatments (9 a.m. on the next 2 days). Body weight and blood glucose level measurement, along with blood sample collection, were performed at 12 p.m. on each day.

**Blood analysis**. Blood glucose levels were determined using glucometers (Bayer Corp., Pittsburgh, PA). Plasma ALT activity was measured using an ALT reagent set (Pointe Scientific Inc., Canton, MI).

**Hematoxylin and eosin (H&E) staining**. Liver paraffin sections were stained with H&E.

**ROS assay**. Liver samples were homogenized in a lysis buffer and mixed with a dichlorofluorescein diacetate fluorescent probe (DCF, Sigma, D6883) to a final probe concentration of 5 μM for 1 h at 37 °C. DCF fluorescence was measured using a BioTek Synergy 2 Multi-Mode Microplate Reader (485 nm excitation and 527 nm emission).

**Reporting summary**. Further information on research design is available in the Nature Research Reporting Summary linked to this article.

## Data availability

The PDB coordination files of compounds DI-1548, DI-1859, 4, 8, 9, and 10 in a complex with DCN1 are deposited in the Protein Data Bank with accession codes 6L, 6O, 6XOQ, 6XOM, 6XON, and 6XOP, respectively. Source Data are provided with this paper.

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

## Acknowledgements

This work was supported in part by National Key R&D Program of China (2016YFA0501800 to Y.S.). Use of the Advanced Photon Source, an Office of Science User Facility operated for the U.S. Department of Energy (DOE) Office of Science by Argonne National Laboratory, was supported by the U.S. DOE under Contract No. DE-AC02-06CH11357. Use of the LS-CAT Sector 21 was supported by the Michigan Economic Development Corporation and the Michigan Technology Tri-Corridor (Grant 085P1000817). The authors thank Dr. Venkatesha Basrur at Proteomics Resource Facility in the University of Michigan for LC–tandem mass spectrometry data support.

## Author contributions

H.Z. designed and synthesized compounds, analyzed data, and wrote the manuscript. J.L. designed and performed cell biology experiments, analyzed data, and wrote the manuscript. L.L. designed and performed the biochemical and biophysical experiments, analyzed data, and wrote the manuscript. D.M. designed and performed in vivo experiments, analyzed data, and wrote the manuscript. C.-Y.Y. and D.B. contributed to computational structure-based design. J.S. designed and performed experiments related to protein expression, X-ray crystallography and mass spectrometric analysis of proteins, analyzed data, and wrote the manuscript. K.C. performed experiments related to protein expression and mass-spectrometric analysis of proteins and analyzed data. L.R. designed and supervised in vivo experiments related to APAP-induced liver injury, analyzed data, and wrote the manuscript. H.S. designed and performed in vivo experiments related to APAP-induced liver injury, analyzed data, and wrote the manuscript. Y.S. contributed to ideas and revision of the manuscript. S.W. initiated and supervised the project, designed experiments, analyzed data, and wrote the manuscript.

## Competing interests

The authors declare no competing interests.
