## [Peer Review File · Nature Communications]

REVIEWER COMMENTS

Reviewer #1 (Remarks to the Author):

The manuscript by Zhou et al describes the development of a highly selective CRL inhibitor for DCN1 that was designed to operate through a covalent mechanism. The mechanism of action is supported through MS and crystallographic studies and shows efficacy in mouse models. Importantly, the compound appears to show a low toxicity profile in mice. The manuscript is logically organized, and of broad interest to the scientific community. The manuscript can be published following the minor suggestions:

1) Page 6: Specifically, DI-1548 and DI-1859 inhibit cullin-3 neddylation at concentrations as low as 1 nM and are thus ~1000-times more potent than DI-591.

From Figure 1, undoubtedly the inhibitors 6 and 7 are multiple-fold more selective than their predecessors. However, the comparison between the different Western blots to claim 1000-fold selectivity may or may not be valid. Densitometry analysis on the same Western would be more appropriate to provide a quantitative estimate of the increase in potency. This can be included in Supplementary information.

2) Figure 2: The data can be presented in a more meaningful manner. It would be helpful to include the structure of the R-(electrophilic)-group above/below each of the compound, to avoid having to re-check Figure 1. The "min" under every label should be removed and stated only once in the axis. The labels are also very small and difficult to read. The colour pallet is also unusual, and difficult to read (for example, the cyan for compound 4 is barely visible). Also note, some graphs in the manuscript have titles, while others do not. These suggestions can also be applied to Figure 4 (labels, size, etc.).

3) The mass spectrometric data is interesting and well developed. One significant concern that is often established with covalent drugs is off-target Cys-interactions. However, from the MS and crystallographic data, it does not seem to be observed under these experimental conditions, which is consistent with the nucleophilicity of the active site Cys and rationale-drug design approach. At the section related to the tryptic digest of DCN3 (page 9 or page 22 respectively) it would be useful to include the number of Cys present in DCN1 protein and a short discussion on absence of any off-target modifications. Is it established whether these covalent molecules are reactive towards glutathione?

4) The in vivo experiments highlight the therapeutic potential of the lead compounds. It would have also been interesting to benchmark these results to MLN4924 to compare effects on neddylation, NRF2, etc.

5) All experiments/Westerns/etc. need to indicate the number of independent replicates (n=?) carried out in the Figure captions. This was stated to be completed in the reporting summary, but this information could not be located in the methodology section or supplementary files.

6) Acronyms and short-forms should be appropriately defined. For example, in Figure 1, what does UT refer to? (untreated?). IUPAC codes should be followed, hrs -> h; mins -> min; etc. There are several places in the manuscript where Cys, CYS, C, cysteine, CYS 115, and CYS115 are all used. This should be standardized.

Minor Points:

Figure 3: Re-word “whose omit map showing lack of covalent bonding.”

Page 13: Re-word “However, the electron withdrawing effect of the protonated imidazole group indeed expedite the Michael addition of compound 10 and DCN1 protein, which make compound 10 has faster reactivity than compound 8 and 9.”

Reviewer #2 (Remarks to the Author):

This manuscript by Zhou et al. describes potent and irreversible covalent inhibitors of the DCN1/UBC12 interaction. There is significant interest, from a therapeutic perspective, in inhibitors of protein-protein interactions, neddylation, and Cullins, so the publication is likely to be of use to drug discovery researchers. The work is based on a publication from Dr. Wang’s lab in 2017 that described DI-591, which showed poor potency. The current work uses a structure-guided approach to enhance the potency of DI-591, and the results are impressive: nanomolar potencies, elucidation of the biological mechanism, an inactive stereoisomer control compound (in keeping with best practices in chemical probe development), demonstration of in vivo activity in both prevention and therapeutic models of acetaminophen-induced liver injury, and six co-crystal structures. The manuscript is

relatively well-written, and the supplementary information and experimental sections appear complete. Although there are other questions the authors could ask (e.g., what happens to the levels of KEAP1 and p62?), I don't believe that these questions should hold up this publication and that they can be addressed in future publications.

I have a few suggestions the authors may consider before publication.

1. It would be helpful to the reader to have a cartoon schematic that depicts the position of the DCN1/UBC12 interaction in the signaling pathway. Because the authors investigate levels of, for instance, HO-1, it would be useful if the schematic were to extend all the way through NRF2 target genes.
2. The pharmacokinetic parameters of the compounds have not been reported. In the absence of these data, I think it would be worthwhile discussing the relationship between pharmacokinetic and pharmacodynamic properties when using an irreversible inhibitor. The discussion should include what is known about the kinetics of resynthesis of DCN1.
3. A number of KEAP1/NRF2 inhibitors have been described. The mechanism described in this publication would be similar in some regards; for instance, an outcome that would be common to both approaches is NRF2 activation, which the authors have demonstrated. Because the authors are focused on a target that is upstream of NRF2/KEAP1, there would likely be other consequences, as well. It would be helpful to discuss the differences between these two strategies (NRF2/KEAP1 and DCN1/UBC12 inhibition).
4. On p. 12, the authors refer to a "concerted" cleavage. I don't believe they have done any experiments to determine whether the mechanism of the C-N cleavage is concerted or stepwise, so I would not make this claim.
5. In Figure 6, the key for G1-G8 is buried within the legend, which makes the figure challenging to understand. Is it possible to include the key directly in the figure?
6. The term "mass spectroscopic" is used in several places in the manuscript and should be replaced with "mass spectrometric."
7. There may have been a cut/paste error in Supplementary Fig. 2; the top and bottom panels appear to be identical.

Reviewed by Terry W. Moore

Reviewer #3 (Remarks to the Author):

1) Summary

The paper by Zhou et al. describes elegant medicinal chemistry work that results in the discovery of two covalent small molecule inhibitors (DI-1548 and DU-1859) of DCN1, a co-E3 ligase that participates in the modification of cullin proteins by Nedd8. The covalent linkage of these compounds to DCN1 has been unequivocally demonstrated by mass spectrometric analysis and co-crystallization. The rapid binding/action of these molecules to DCN1 was revealed by mass spectrometric analysis, Bio-layer Interferometry, and cell-based experiments. Remarkably, DI-1548 and DI-1859 exhibit selective inhibition of cullin 3 neddylation to elevate the NRF2 protein in cells at sub-nM levels, a potency that is ~1,000 times greater than the previous non-covalent DCN1 inhibitor DI-591. In addition, the robust response of DI-1859 in the stabilization of NRF2 was observed in mice, which contributed to protection of mice from acetaminophen-induced liver damage.

2) Critique

a) Significance: This work is of outstanding significance because it has created two highly useful tool compounds DI-1548 and DU-1859 that selectively inhibit the activity of Cullin 3-RING E3 Ubiquitin Ligases (CRL3s) with sub-nM potency. These molecules are expected to be highly valuable in proving the biological function of CRL3s. It may have transformative impact in the field of ubiquitin-dependent protein degradation.

b) Novelty: DI-1548 and DI-1859 are the first-in-class covalent small molecule inhibitors of CRL3.

c) Technical merit: The technical quality is extremely high. The mass spectrometric analysis and co-crystallization are first rate. The results are compelling and absolutely convincing.

3) Other comments

While I believe that based on reasons above, this paper in its current form is sufficient for publication at Nature Communication, I would suggest a couple of points of improvement for the authors to consider. First, it would be very helpful if the authors could develop cell-based, NRF2-dependent assays to quantify the effects of DI-1548 and DI-1859. Second, this work would suggest that DCN1 predominantly acts to catalyze cullin 3 neddylation. It would be extremely intriguing if the authors could use structural modeling to provide explanation.

More rigorous English editing may be needed. Page 17: change "...substrates of cullin 1 CRL..." to "...substrates of CRL1..."

Point-by-point response to the reviewers' comments:

Response to the comments of Reviewer #1:

Reviewer #1 (Remarks to the Author):

The manuscript by Zhou et al describes the development of a highly selective CRL inhibitor for DCN1 that was designed to operate through a covalent mechanism. The mechanism of action is supported through MS and crystallographic studies and shows efficacy in mouse models. Importantly, the compound appears to show a low toxicity profile in mice. The manuscript is logically organized, and of broad interest to the scientific community. The manuscript and can be published following the minor suggestions:

1) Page 6: Specifically, DI-1548 and DI-1859 inhibit cullin-3 neddylation at concentrations as low as 1 nM and are thus ~1000-times more potent than DI-591. From Figure 1, undoubtedly the inhibitors 6 and 7 are multiple-fold more selective than their predecessors. However, the comparison between the different Western blots to claim 1000-fold selectivity may or may not be valid. Densitometry analysis on the same Western would be more appropriate to provide a quantitative estimate of the increase in potency. This can be included in Supplementary information.

Response: As we analyzed DI-1548/DI-1859 and DI-591 on the same western blot in Figure 6a and 6b, we carried out the densitometry analysis of neddylation-cullin3 in Figure 6a and 6b to provide quantitative evaluation of the potencies of compounds DI-1548, DI-1859 and DI-591. The result is added as the Supplementary Figure 21 which shows both DI-1548 and DI-1859 are 300-1000 times more potent than DI-591 in inhibition of Cullin 3 neddylation.

2) Figure 2: The data can be presented in a more meaningful manner. It would be helpful to include the structure of the R-(electrophilic)-group above/below each of the compound, to avoid having to re-check Figure 1. The “min” under every label should be removed and stated only once in the axis. The labels are also very small and difficult to read. The colour pallet is also unusual, and difficult to read (for example, the cyan for compound 4 is barely visible). Also note, some graphs in the manuscript have titles, while others do not. These suggestions can also be applied to Figure 4 (labels, size, etc.).

Response: Following these suggestions, the structures of the warhead groups have been included above each compound in Figure 2a, and the labels, size and the color have been revised in both figures.

3) The mass spectrometric data is interesting and well developed. One significant concern that is often established with covalent drugs is off-target Cys-interactions. However, from the MS and crystallographic data, it does not seem to be observed under these experimental conditions, which is consistent with the nucleophilicity of the active site Cys and rationale-drug design approach. At the section related to the tryptic digest or DCN3 (page 9 or page 22 respectively) it would be useful to include the number of Cys present in DCN1 protein and a short discussion on absence of any off-target modifications. Is it established whether these covalent molecules are reactive towards glutathione?

Response: The following revisions have been made:

On page 9, the following is added: “There are three cysteine residues (Cys90, Cys115 and Cys131) in DCN1 protein and Cys115 is the expected one to form covalent bond with these covalent inhibitors.”

On page 14, the following is added: “DCN3 contains six cysteine residues including a conserved Cys140 corresponding to Cys115 in DCN1, and DI-591 binds to DCN3 with a weak affinity ($K_d > 10 \mu\text{M}$)¹.”.

On page 24, the following is added: “although DCN3 have a conserved Cys140 residue in the binding site corresponding to Cys115 in DCN1.”.

On page 25, the discussion of off-target modifications is revised: “Although covalent small-molecule inhibitors have been successfully developed for the treatment of human diseases, including cancer, there is often a concern of their potential toxicity from possible off-target reactivity^{2,3}. We have demonstrated that DI-1548 and DI-1859 selectively and rapidly react to Cys115 in DCN1 and shows no reactivity to DCN3 despite the fact that DCN3 has a conserved Cys140 in its binding site corresponding to Cys115 in DCN1. To further address the major concern associated with a covalent inhibitor *in vivo*, we evaluated the potential toxicity of DI-1859 in mice with daily dosing at 50 mg/kg for 14 consecutive days. Our extensive histology analysis on a large number of tissues, including liver, lung, kidney and heart, showed that DI-1859 causes no tissue damage in any tissue in mice, indicating that DI-1859 has very low toxicity in mice.”.

As we have developed mass-spectrometric assay to test the reactivity and the selectivity of the covalent compounds to proteins such as DCN1 and DCN3, the reactivity of these covalent compounds towards glutathione is not tested. Of note, in our developed mass-spectrometric assay, the covalent DCN1 inhibitors form covalent bond with DCN1 although high concentration

of reducing agent dithiothreitol (DTT, 1 mM) was presented. Both glutathione and dithiothreitol contain sulfhydryl group for the Michael reaction with the warhead of the covalent compounds, which indicate the low reactivity of our compounds towards to glutathione and dithiothreitol.

4) The in vivo experiments highlight the therapeutic potential of the lead compounds. It would have also been interesting to benchmark these results to MLN4924 to compare effects on neddylation, NRF2, etc.

Response: As an inhibitor of NEDD8 activating enzyme (E1 enzyme NAE), MLN4924 broadly inhibits the neddylation of all cullin members and inactivates all CRLs^{4,5}. Consistent with the global accumulation of CRL substrates, MLN4924 shows strong cytotoxicity as shown in the Supplementary Fig. 24. However, DI-1548 selectively inhibits CRL3 and shows no cytotoxicity in the tested four cancer cell lines at concentrations up to 1,000 nM (**Supplementary Fig. 24**). Based on robust NRF2 activation, coupled with its lack of cytotoxicity, we performed the in vivo experiments for DI-1859 on the evaluation of mitigating APAP-induced hepatotoxicity in mice. Because of the strong cytotoxicity, we did not include MLN4924 in these experiments.

5) All experiments/Westerns/etc. need to indicate the number of independent replicates (n=?) carried out in the Figure captions. This was stated to be completed in the reporting summary, but this information could not be located in the methodology section or supplementary files.

Response: The number of independent replicates for the experiments have been included in the Figure captions.

6) Acronyms and short-forms should be appropriately defined. For example, in Figure 1, what does UT refer to? (untreated?). IUPAC codes should be followed, hrs -> h; mins -> min; etc. There are several places in the manuscript where Cys, CYS, C, cysteine, CYS 115, and CYS115 are all used. This should be standardized.

Response: Revised based on the comments.

“UT” has been changed to “DMSO” in Figures 1 and 5, Supplementary Figures 18, 19 and 20.

“h” and “min” are used throughout the manuscript.

Cys115 is used throughout the manuscript.

Minor Points:

Figure 3: Re-word “whose omit map showing lack of covalent bonding.”

Page 13: Re-word “However, the electron withdrawing effect of the protonated imidazole group indeed expedite the Michael addition of compound 10 and DCN1 protein, which make compound 10 has faster reactivity than compound 8 and 9.”

Response: The revisions have been made as following:

“whose omit map showing lack of covalent bonding.” changed to “whose omit map shows the lack of covalent bonding.”.

“However, the electron withdrawing effect of the protonated imidazole group indeed expedite the Michael addition of compound 10 and DCN1 protein, which make compound 10 has faster reactivity than compound 8 and 9.” changed to “However, compound **10** still has a faster

reactivity to DCN1 than compounds **8** and **9** probably due to the electron withdrawing effect of the protonated imidazole group.”.

Response to the comments of Reviewer #2:

Reviewer #2 (Remarks to the Author):

This manuscript by Zhou et al. describes potent and irreversible covalent inhibitors of the DCN1/UBC12 interaction. There is significant interest, from a therapeutic perspective, in inhibitors of protein-protein interactions, neddylation, and Cullins, so the publication is likely to be of use to drug discovery researchers. The work is based on a publication from Dr. Wang's lab in 2017 that described DI-591, which showed poor potency. The current work uses a structure-guided approach to enhance the potency of DI-591, and the results are impressive: nanomolar potencies, elucidation of the biological mechanism, an inactive stereoisomer control compound (in keeping with best practices in chemical probe development), demonstration of in vivo activity in both prevention and therapeutic models of acetaminophen-induced liver injury, and six co-crystal structures. The manuscript is relatively well-written, and the supplementary information and experimental sections appear complete. Although there are other questions the authors could ask (e.g., what happens to the levels of KEAP1 and p62?), I don't believe that these questions should hold up this publication and that they can be addressed in future publications.

I have a few suggestions the authors may consider before publication.

Response: The effect of DI-1548 and DI-1859 on the protein levels of KEAP1 and p62 are evaluated in both U2OS cell line and THLE2 liver cells, and the result is added as Supplementary Figure 22. No obvious effect was observed on both KEAP1 and p62 and the following is added in the main text:

“As NRF2 is negatively controlled by KEAP1 and p62 is able to active NRF2 by disrupting the association of NRF2 and KEAP1, we thus examined the effect of our DCN1 inhibitor on the

protein levels of KEAP1 and p62. DI-1548 and DI-1859 have no obvious effect on the protein level of KEAP1 and p62 in U2OS cell line and THLE2 liver cells (**Supplementary Fig. 22**), which indicates that the regulation of NRF2 by DCN1 inhibitors is via the inhibition of cullin 3 neddylation.”

1. It would be helpful to the reader to have a cartoon schematic that depicts the position of the DCN1/UBC12 interaction in the signaling pathway. Because the authors investigate levels of, for instance, HO-1, it would be useful if the schematic were to extend all the way through NRF2 target genes.

Response: A cartoon schematic is added as Figure 8 and the following description has also been added as follow.

Figure 8. Proposed mechanism of action of covalent DCN1 inhibitor DI-1548 for its protection of APAP-induced liver damage via selective inhibition of cullin 3 neddylation. DI-1548 covalently binds to DCN1 and disrupts the interaction of DCN1-UBC12 and causes the collapse of neddylation complex. The resultant inactive CRL3

leads to the accumulation of NRF2 which upregulates its targeted genes and effectively protects mice from acetaminophen-induced liver damage.

Figure 8 illustrates the mechanism of action for our covalent DCN1 inhibitors. DCN1 functions as a scaffold protein and upholds the neddylation complex (RBX1-UBC12~NEDD8-CUL3-DCN1) through interacting with both UBC12 and cullin3 proteins. Upon neddylation, cullin3 undergoes a conformational change to promote the assembly and activation of the CRL3 complex (RBX1-CUL3-Keap1)⁶. This activated CRL3 complex recruits substrate proteins, such as NRF2, leading to their ubiquitination, followed by degradation by the 26S proteasome.

Covalent inhibitors such as DI-1548 and DI-1859, form a covalent bond with DCN1 and disrupt the interaction of DCN1 with UBC12, leading to dissociation and inactivation of the CRL3 complex. The inactivation of the CRL3 complex results in accumulation of CRL3 substrate proteins, such as NRF2 in cells and tissues. Accumulated NRF2 protein is translocated into the nucleus and regulates its targeted genes including *HO-1* and *NQO1*, which protect mice from liver damage induced by acetaminophen.

2. The pharmacokinetic parameters of the compounds have not been reported. In the absence of these data, I think it would be worthwhile discussing the relationship between pharmacokinetic and pharmacodynamic properties when using an irreversible inhibitor. The discussion should include what is known about the kinetics of resynthesis of DCN1.

Response: Pharmacokinetics (PK) analysis of DI-1859 in mice plasma, liver, kidney, lung, heart and brain are added as Supplementary Table 2.

On page 19, the following is added: “Of note, the PD effect for DI-1859 in liver tissue extended after the compound was cleared from systemic circulation (< 6 h) based upon the pharmacokinetics (PK) analysis (**Supplementary Table 2**).”

On page 26, a short discussion is added: “A potential major advantage of covalent inhibition is the persistent effect on its biological target as compared to non-covalent inhibition⁷. Indeed, DI-1548 and DI-1859 not only achieve >300-times better cellular potency than their noncovalent counterparts, they also deliver persistent PD effect, even after the drugs are cleared from plasma.”.

3. A number of KEAP1/NRF2 inhibitors have been described. The mechanism described in this publication would be similar in some regards; for instance, an outcome that would be common to both approaches is NRF2 activation, which the authors have demonstrated. Because the authors are focused on a target that is upstream of NRF2/KEAP1, there would likely be other consequences, as well. It would be helpful to discuss the differences between these two strategies (NRF2/KEAP1 and DCN1/UBC12 inhibition).

Response: We have proposed a schematic (Fig. 8) to illustrate the mechanism of action of inhibition of DCN1/UBC12 and NRF2/KEAP1. The following discussion have been added:

Also illustrated in Figure 8, KEAP1 functions as the substrate adaptor in the CRL3 complex through interacting with cullin3 and NRF2. Hence, small-molecule inhibitors of the KEAP1-NRF2 interaction are also capable of induce the accumulation of NRF2^{8,9}. Thus, pharmacological activation of NFR2 can be achieved by both DCN1/UBC12 inhibitors and KEAP1/NRF2 inhibitors. Our data indicate that DCN1/UBC12 inhibitors selectively inhibit the neddylation of

cullin3 and subsequently inactivate CRL3, which induces the accumulation of CRL3 substrates including but not limited to NRF2. In comparison, KEAP1/NRF2 inhibitors block the interactions of KEAP1 with NRF2 or potentially other KEAP1 interacting proteins, which use the same docking site, leading to upregulations of those proteins recruited by KEAP1 to CRL3 for ubiquitination. Additional studies are required to understand the similarities and differences on regulation of substrate proteins between DCN1/UBC12 inhibitors and KEAP1/NRF2 inhibitors.

4. On p. 12, the authors refer to a “concerted” cleavage. I don’t believe they have done any experiments to determine whether the mechanism of the C-N cleavage is concerted or stepwise, so I would not make this claim.

Response: Based upon the striking different reaction kinetics of compounds 2-10 and DCN1 protein (as shown in mass spectrometric data) and the co-crystal structures of these compounds complexed with DCN1 protein, we proposed a reaction mechanism for the formal SN2’ substitution reaction between DCN1 with DI-1548 or with DI-1859. However, no experiment was done to determine whether the C-S formation and C-N cleavage are concerted or stepwise. Therefore, the “concerted” on p. 12 has been deleted.

5. In Figure 6, the key for G1-G8 is buried within the legend, which makes the figure challenging to understand. Is it possible to include the key directly in the figure?

Response: The description of G1-G8 have been included in the figure as shown in the revised Figure 7 (page 21).

6. The term “mass spectroscopic” is used in several places in the manuscript and should be replaced with “mass spectrometric.”

Response: “mass spectroscopic” was changed to “mass spectrometric” throughout the manuscript.

7. There may have been a cut/paste error in Supplementary Fig. 2; the top and bottom panels appear to be identical.

Response: Thanks for finding this error and this has been corrected.

Response to the comments of Reviewer #3:

Reviewer #3 (Remarks to the Author):

1) Summary

The paper by Zhou et al. describes elegant medicinal chemistry work that results in the discovery of two covalent small molecule inhibitors (DI-1548 and DU-1859) of DCN1, a co-E3 ligase that participates in the modification of cullin proteins by Nedd8. The covalent linkage of these compounds to DCN1 has been unequivocally demonstrated by mass spectrometric analysis and co-crystallization. The rapid binding/action of these molecules to DCN1 was revealed by mass spectrometric analysis, Bio-layer Interferometry, and cell-based experiments. Remarkably, DI-1548 and DI-1859 exhibit selective inhibition of cullin 3 neddylation to elevate the NRF2 protein in cells at sub-nM levels, a potency that is ~1,000 times greater than the previous non-covalent DCN1 inhibitor DI-591. In addition, the robust response of DI-1859 in the stabilization of NRF2 was observed in mice, which contributed to protection of mice from acetaminophen-induced liver damage.

2) Critique

a) Significance: This work is of outstanding significance because it has created two highly useful tool compounds DI-1548 and DU-1859 that selectively inhibit the activity of Cullin 3-RING E3 Ubiquitin Ligases (CRL3s) with sub-nM potency. These molecules are expected to be highly valuable in proving the biological function of CRL3s. It may have transformative impact in the field of ubiquitin-dependent protein degradation.

b) Novelty: DI-1548 and DI-1859 are the first-in-class covalent small molecule inhibitors of CRL3.

c) Technical merit: The technical quality is extremely high. The mass spectrometric analysis and co-crystallization are first rate. The results are compelling and absolutely convincing.

3) Other comments

While I believe that based on reasons above, this paper in its current form is sufficient for publication at Nature Communication, I would suggest a couple of points of improvement for the authors to consider. First, it would be very helpful if the authors could develop cell-based, NRF2-dependent assays to quantify the effects of DI-1548 and DI-1859. Second, this work would suggest that DCN1 predominantly acts to catalyze cullin 3 neddylation. It would be extremely intriguing if the authors could use structural modeling to provide explanation.

Response: The mRNA levels of NRF2 target genes *NQO1* and *HO1* are examined in the THLE2 liver cells by qRT-PCR and the result is added as Supplementary Figure 23. DI-1859 increases the mRNA levels of NQO1 and HO1 by 17 and 3 folds, respectively (Supplementary Fig. 23). The qRT-PCR assay is a suitable assay to quantify the effects of DI-1548 and DI-1859.

More rigorous English editing may be needed. Page 17: change "...substrates of cullin 1 CRL..." to "...substrates of CRL1..."

Response: Revised accordingly throughout the manuscript.

References:

1. Zhou, H., *et al.* A potent small-molecule inhibitor of the DCN1-UBC12 interaction that selectively blocks cullin 3 neddylation. *Nat Commun* **8**, 1150 (2017).
2. Singh, J., Petter, R.C., Baillie, T.A. & Whitty, A. The resurgence of covalent drugs. *Nat. Rev. Drug Discov.* **10**, 307-317 (2011).
3. Mah, R., Thomas, J.R. & Shafer, C.M. Drug discovery considerations in the development of covalent inhibitors. *Biorg. Med. Chem. Lett.* **24**, 33-39 (2014).
4. Soucy, T.A., *et al.* An inhibitor of NEDD8-activating enzyme as a new approach to treat cancer. *Nature* **458**, 732-U767 (2009).
5. Brownell, J.E., *et al.* Substrate-Assisted Inhibition of Ubiquitin-like Protein-Activating Enzymes: The NEDD8 E1 Inhibitor MLN4924 Forms a NEDD8-AMP Mimetic In Situ. *Molecular Cell* **37**, 102-111 (2010).
6. Duda, D.M., *et al.* Structural insights into NEDD8 activation of cullin-RING ligases: conformational control of conjugation. *Cell* **134**, 995-1006 (2008).
7. Baillie, T.A. Targeted Covalent Inhibitors for Drug Design. *Angew. Chem. Int. Edit.* **55**, 13408-13421 (2016).
8. Rojo de la Vega, M., Chapman, E. & Zhang, D.D. NRF2 and the Hallmarks of Cancer. *Cancer Cell* **34**, 21-43 (2018).
9. Zhou, H., Wang, Y., You, Q. & Jiang, Z. Recent progress in the development of small molecule Nrf2 activators: a patent review (2017-present). *Expert Opin Ther Pat* **30**, 209-225 (2020).

REVIEWERS' COMMENTS

Reviewer #2 (Remarks to the Author):

The authors have sufficiently addressed my concerns.